# StelLA: Subspace Learning in Low-rank Adaptation using Stiefel Manifold

**Zhizhong Li, Sina Sajadmanesh, Jingtao Li, Lingjuan Lyu**[*]
Sony AI
Zurich, Switzerland
{zhizhong.li,sina.sajadmanesh,jingtao.li,lingjuan.lv}@sony.com

## Abstract

Low-rank adaptation (LoRA) has been widely adopted as a parameter-efficient technique for fine-tuning large-scale pre-trained models. However, it still lags behind full fine-tuning in performance, partly due to its insufficient exploitation of the geometric structure underlying low-rank manifolds. In this paper, we propose a geometry-aware extension of LoRA that uses a three-factor decomposition $USV^\top$. Analogous to the structure of singular value decomposition (SVD), it separates the adapter's input and output subspaces, $V$ and $U$, from the scaling factor $S$. Our method constrains $U$ and $V$ to lie on the Stiefel manifold, ensuring their orthonormality throughout the training. To optimize on the Stiefel manifold, we employ a flexible and modular geometric optimization design that converts any Euclidean optimizer to a Riemannian one. It enables efficient subspace learning while remaining compatible with existing fine-tuning pipelines. Empirical results across a wide range of downstream tasks, including commonsense reasoning, math and code generation, image classification, and image generation, demonstrate the superior performance of our approach against the recent state-of-the-art variants of LoRA. Code is available at `https://github.com/SonyResearch/stella`.

## 1 Introduction

The rise of large foundation models [2, 21] has transformed machine learning, driving breakthroughs across a diverse range of applications [59, 65, 41]. These models, often with billions of parameters, show exceptional performance in fields such as natural language understanding [22], computer vision [76], and multi-modal learning [67]. Yet, their substantial scale results in massive computational and storage costs, limiting broader adoption via task-specific fine-tuning [29].

To address this challenge, parameter-efficient fine-tuning (PEFT) methods, such as prefix tuning [35], prompt tuning [50], and adapter tuning [28, 29], have gained considerable attention. Among these, Low-Rank Adaptation (LoRA) [29] is widely adopted due to its ability to efficiently adapt pre-trained models to downstream tasks without altering the original network architecture or incurring extra inference costs. LoRA operates by learning low-rank adaptations to a selected set of linear layers while keeping the original weights frozen. Given a pre-trained weight matrix $W \in \mathbb{R}^{m \times n}$, LoRA computes the adapted weight as $W + BA^\top$, where $B \in \mathbb{R}^{m \times r}$ and $A \in \mathbb{R}^{n \times r}$. Choosing a small $r \ll \min(m, n)$ significantly reduces the number of trainable parameters. Nevertheless, a performance gap often exists between LoRA and full fine-tuning due to the limited capacity of the low-rank matrices $A$ and $B$ to capture complex updates required for optimal task performance [23].

A common strategy to improve LoRA's performance is to guide its initialization by leveraging structural insights from the pre-trained model [73]. Particularly, singular value decomposition

---

[*]Corresponding author

39th Conference on Neural Information Processing Systems (NeurIPS 2025).

(SVD) has been used for uncovering informative subspaces in weight matrices, activations, or gradients [43, 64, 46]. SVD factorizes a given matrix $M$ into $M = U\Sigma V^\top$, where the columns of $U$ and $V$ form orthonormal bases for the output and input spaces of $M$, respectively, ordered by the importance dictated by the singular values in $\Sigma$. This property makes SVD well-suited for identifying low-dimensional subspaces that preserve meaningful information from the pre-trained model.

While SVD-based initialization methods for LoRA have shown promise, their influence is limited to the start of training, offering minimal guidance for the subsequent optimization process. This naturally leads to the question: *whether explicitly optimizing the subspace throughout training can yield further performance improvements?* Moreover, the existence of various heuristics for subspace selection—such as focusing on leading *vs.* trailing components [43, 63], or considering weights *vs.* gradients [43, 64]—suggests that intuition-driven manual subspace selection is suboptimal. This motivates a more principled approach to learn subspaces from data throughout training.

To bridge this gap, we propose a novel approach that directly optimizes the input-output subspaces of LoRA during training. Our key insight is to mirror the structure of the truncated SVD by representing the low-rank adaptation for weight matrix $W$ as a three-factor formulation,

$$W + USV^\top, \tag{1}$$

where $U \in \mathbb{R}^{m \times r}$ and $V \in \mathbb{R}^{n \times r}$ define the orthonormal bases for the output and input subspaces, respectively, and $S \in \mathbb{R}^{r \times r}$ captures the transformation between them. To ensure that the subspace parameters $U$ and $V$ remain orthonormal during training, we constrain them to lie on the *Stiefel manifold*—the set of matrices with orthonormal columns. This leads to our method, **St**ie**f**el **L**ow-rank **A**daptation (StelLA), which performs LoRA using a subspace-aware formulation. By leveraging the geometric optimization on the Stiefel manifold, StelLA maintains the geometric structure of the subspaces in training, allowing for principled and effective learning of low-rank adaptation.

Recently, methods such as DoRA [40] found that decomposing the weight into magnitude and direction components can improve LoRA's performance. This aligns with our approach, as the Stiefel manifold constraint on $U$ and $V$ captures the direction, and $S$ models the magnitude. Besides, maintaining the orthogonality of low-rank matrices $U$ and $V$ during training is also beneficial for certain downstream analyses, such as adversarial robustness [55, 57].

We benchmark StelLA across a diverse set of domains, including natural language understanding, natural language generation, visual understanding, and visual generation. Compared with the state-of-the-art LoRA variants, StelLA consistently achieves superior performance across all evaluated tasks. Relative to the strongest baseline, StelLA achieves notable improvements: up to +1.3 accuracy points on commonsense reasoning, +2.33 on math and code generation, +0.25 on image classification, and a 7.11 point reduction in FID for text-to-image generation.

Our contributions are summarized as follows: (1) We propose a novel three-factor representation for LoRA incorporating Stiefel manifold constraints, enabling the optimization of LoRA's input and output subspaces directly during training. (2) We present a flexible geometric optimization algorithm for the Stiefel manifold, allowing seamless integration with existing Euclidean optimizers. (3) We conduct ablation studies on the design choices and evaluate the impact of different geometric structures. (4) We verify the effectiveness of StelLA across a variety of tasks and models, encompassing natural language understanding and generation, image classification, and text-to-image generation.

## 2 Related Work

**LoRA Initialization Methods and the Use of SVD.** Several recent works explore improved LoRA initialization using matrix decomposition techniques like SVD to better align the adaptation subspace with pretrained weights. PiSSA [43] and LaMDA [4] initialize adapters using the leading $r$ singular vectors from SVD of the pretrained weights, while MiLoRA [63] uses the trailing $r$ singular vectors. EVA [46] applies SVD to activations and adapts rank based on the spectrum, while LoRA-GA [64] uses SVD on gradients to align with task-relevant directions. Beyond SVD, OLoRA [10] employs QR decomposition and selects the first $r$ columns from the orthonormal matrix. These approaches exploit meaningful subspaces to improve initialization, whereas ours directly learns the optimal subspace during training, allowing for more flexible and effective task-specific adaptation.

**Geometric Constraints for LoRA.** Zhang and Pilanci [72] use Riemannian geometry to precondition the LoRA optimization, whereas we optimize the adapter's subspaces using the Stiefel manifold. OFT [51] learns orthonormal rotations of pretrained weights via Cayley parameterization. Spectral Adapter [73] fine-tunes within the top spectral subspace, with a variant that rotates leading singular vectors using orthonormal matrices. In contrast, our approach enforces orthogonality directly on the input and output subspace projections via Stiefel manifold optimization, offering a more flexible and expressive adaptation mechanism. Concurrently, PoLAR [39] also sets $U$ and $V$ on Stiefel manifold, but it uses a landing algorithm with infeasibility penalty instead of retraction for optimization.

**LoRA with Three Factors.** TriLoRA [20] and MoSLoRA [66] use the three-factor formulation for LoRA, but they do not keep the orthogonality of the two projections during training. LoRA-XS [5] also uses the three-factor formulation, but they freeze the two projections and only train the middle $r \times r$ matrix for extreme efficiency. Both AdaLoRA [74] and GeoLoRA [58] adopt a three-factor formulation, with $U$ and $V$ constrained to be orthogonal matrices. AdaLoRA achieves orthogonality via regularization, while GeoLoRA uses gradient flow. However, their focus is to achieve rank adaptability by inspecting the singular values of $S$. StelLA can be readily combined with AdaLoRA's rank adaptation strategy by constraining $S$ to be diagonal, which we leave to future work.

**Other LoRA Variants.** rsLoRA [30] introduces a scaling factor based on the square root of the rank for better stability. LoRA+ [24] accelerates convergence by applying separate learning rates to the two matrices. DoRA [40] and DeLoRA [7] decouple LoRA updates into magnitude and direction. ReLoRA [38] periodically merges adapter weights to improve expressiveness. QLoRA [15] integrates quantization with LoRA to minimize memory usage. These efforts are orthogonal to our work. Besides LoRA, there are other adaptation methods like Prefix-Tuning [36], Prompt Tuning [33], Adapters [28], and others [17], which we do not cover in this paper.

## 3 Preliminaries

Here we briefly review the related concepts in differential geometry and geometric optimization. For a more detailed introduction, we refer the reader to Edelman et al. [18], Absil et al. [1], Li et al. [37], Roy et al. [54], Becigneul and Ganea [6]. We also prepared an intuitive example in low dimensions in Appendix A to help readers understand the geometric concepts. The Stiefel manifold, denoted $\mathrm{St}(k, n)$, is the set of all $n \times k$ matrices with orthonormal columns, *i.e.*, $\mathrm{St}(k, n) = \{Y \in \mathbb{R}^{n \times k} \mid Y^\top Y = I_k\}$. To optimize a function $f : \mathrm{St}(k, n) \to \mathbb{R}$, we require tools from the Riemannian geometry. The tangent space at a point $Y \in \mathrm{St}(k, n)$, denoted $T_Y \mathrm{St}(k, n)$, consists of matrices $\Delta \in \mathbb{R}^{n \times k}$ satisfying $Y^\top \Delta + \Delta^\top Y = 0$. A Riemannian metric defines an inner product on the tangent space, and the canonical metric in Stiefel manifold is $g_Y(\Delta_1, \Delta_2) = \mathrm{tr}(\Delta_1^\top (I_n - \frac{1}{2} Y Y^\top) \Delta_2)$. The Riemannian gradient $\mathrm{grad}_Y$ w.r.t. function $f$ can be computed from the Euclidean gradient $\nabla_Y$ by

$$\mathrm{grad}_Y = \nabla_Y - Y \nabla_Y^\top Y. \tag{2}$$

If the Riemannian gradient is modified, *e.g.*, by adding momentum, it may no longer lie in the tangent space of the manifold. To perform geometric optimization, we need to project it back to the tangent space using the projection $\pi_Y : T_Y \mathbb{R}^{n \times k} \to T_Y \mathrm{St}(k, n)$,

$$\pi_Y(\Delta) = \Delta - Y \mathrm{symm}(Y^\top \Delta), \tag{3}$$

where $\mathrm{symm}(A) = \frac{1}{2}(A + A^\top)$ symmetrizes a matrix. After taking a step along a tangent vector $\Delta$, the resulting point $Y + \Delta$ typically leaves the manifold. Hence, a retraction is used to map it back. In this work, we use the polar retraction, defined as

$$\rho_Y(\Delta) = \mathrm{uf}(Y + \Delta), \tag{4}$$

where $\mathrm{uf}(.)$ returns the orthogonal matrix in the polar decomposition.

## 4 Subspace Learning in LoRA Using Stiefel Manifold

We now present Stiefel Low-rank Adaptation (StelLA), which learns the input and output subspaces of the adapter directly during fine-tuning. Given a pre-trained weight matrix $W \in \mathbb{R}^{m \times n}$, the goal is to fit it to a downstream task using a three-factor low-rank adapter,

$$\tilde{W} = W + \frac{\alpha}{r} U S V^\top, \tag{5}$$

---

**Algorithm 1** StelLA: Stiefel Low-Rank Adaptation

---

**Require:** Pre-trained weight $W \in \mathbb{R}^{m \times n}$, loss function $\mathcal{L}$, a Euclidean optimizer's step function 'step', rank $r$, scale factor $\alpha$, number of iterations $T$.
1: Randomly initialize $U_0 \in \mathrm{St}(r, m)$ and $V_0 \in \mathrm{St}(r, n)$, set $S_0 \leftarrow I_r$.
2: **for** $t \leftarrow 0$ to $T - 1$ **do**
3:     Compute loss: $\mathcal{L}_t \leftarrow \mathcal{L}(W + \frac{\alpha}{r} U_t S_t V_t^\top)$.
4:     Compute Euclidean gradients: $\nabla_{U_t}, \nabla_{S_t}, \nabla_{V_t}$.              ▷ *via automatic differentiation*
5:     Convert Euclidean gradients to Riemannian:                      ▷ Equation (2)

$$\mathrm{grad}_{U_t} \leftarrow \nabla_{U_t} - U_t \nabla_{U_t}^\top U_t, \quad \mathrm{grad}_{V_t} \leftarrow \nabla_{V_t} - V_t \nabla_{V_t}^\top V_t.$$

6:     Take tentative steps using the given optimizer's step function:           ▷ *e.g.*, using Adam

$$\tilde{U}_{t+1} \leftarrow \mathrm{step}(U_t, \mathrm{grad}_{U_t}), \quad \tilde{V}_{t+1} \leftarrow \mathrm{step}(V_t, \mathrm{grad}_{V_t}), \quad S_{t+1} \leftarrow \mathrm{step}(S_t, \nabla_{S_t}).$$

7:     Project the perturbed gradients $\tilde{U}_{t+1} - U_t$, $\tilde{V}_{t+1} - V_t$ back to the tangent space:    ▷ Equation (3)

$$\Delta_{U_t} \leftarrow \pi_{U_t}(\tilde{U}_{t+1} - U_t), \quad \Delta_{V_t} \leftarrow \pi_{V_t}(\tilde{V}_{t+1} - V_t).$$

8:     Update and retract back to the manifold: $U_{t+1} \leftarrow \rho_{U_t}(\Delta_{U_t})$, $V_{t+1} \leftarrow \rho_{V_t}(\Delta_{V_t})$.     ▷ Equation (4)
9: **end for**
10: **return** Adapted weight: $\tilde{W} \leftarrow W + \frac{\alpha}{r} U_T S_T V_T^\top$.

---

where $U \in \mathrm{St}(r, m)$ and $V \in \mathrm{St}(r, n)$ provide orthonormal basis for the output and input subspaces, respectively. $S \in \mathbb{R}^{r \times r}$ learns a mapping, with the rank $r \ll \min(m, n)$. $S$ can also be further constrained to be diagonal to reduce the number of parameters. $\alpha \in \mathbb{R}$ is a scale hyperparameter.

Let $\mathcal{L}$ denote the task-specific loss. We optimize the following objective (for notational brevity, we express the objective for one StelLA layer):

$$\min_{U \in \mathrm{St}(r,m), \ S \in \mathbb{R}^{r \times r}, \ V \in \mathrm{St}(r,n)} \mathcal{L}\left(W + \frac{\alpha}{r} USV^\top\right), \tag{6}$$

leading to a constrained optimization over Stiefel manifolds for $U$ and $V$, and an unconstrained optimization for $S$.

The optimization is carried out using Algorithm 1. The algorithm begins by initializing $U_0$, $V_0$, and $S_0$ (line 1). At training iteration $t$, the loss is computed based on the adapted weight in the forward pass (line 3), and Euclidean gradients with respect to $U_t$, $S_t$, and $V_t$ are obtained (line 4) in the backward pass using standard automatic differentiation in deep learning frameworks. Since $U_t$ and $V_t$ must remain on the Stiefel manifold, their gradients are converted to Riemannian gradients using Equation (2) (line 5). These gradients, along with the gradient of $S_t$, are passed to an existing Euclidean optimizer's step function, *e.g.*, using SGD, Adam, or RMSProp, to generate tentative updates (line 6). The perturbed gradient by the optimizer is reverse-engineered by examining the difference $\tilde{U}_{t+1} - U_t$. Since the perturbed gradient may not lie on the tangent space of the manifold, it is projected back to the tangent spaces at the current points (line 7). Then, the update is performed by using the polar retraction (line 8). The algorithm ensures that $U$ and $V$ evolve in the Stiefel manifold throughout training. After $T$ steps, the final adapted weight is returned (line 10).

**Initialization.** Both $U$ and $V$ are initialized with random column-orthonormal matrices [56], and we initialize $S$ with the identity matrix. In prior work [43], when the adapter is initialized with non-zero values, the initialization is subtracted from the pre-trained weight matrix to simulate a zero-initialized adapter. We empirically find this trick to be unnecessary in our case. An in-depth ablation of initialization strategies is studied in Section 5.5.

**Implementation.** Algorithm 1 is designed to be modular and flexible, enabling integration with any existing Euclidean optimizer. The 'step' function abstracts the optimizer's internal logic, such as momentum updates or adaptive learning rates, allowing StelLA to seamlessly integrate with a wide range of optimizers, including SGD, Adam, and others, while cleanly separating geometric constraints from the choice of optimization algorithm. We implement StelLA in PyTorch [48] using optimizer hooks. Specifically, line 5 is implemented as a pre-hook to the optimizer step, while lines 7–8 are implemented as a post-hook. Our implementation is readily integrable with HuggingFace's PEFT library [42], enabling easy adoption by the community.

The polar retraction is computed via SVD, which is the most expensive operation in the algorithm. To improve efficiency, we stack all $U$s and $V$s with identical shapes across different layers and apply a batched SVD. This batched strategy yields 15-20$\times$ speed up in our experiments, effectively eliminating the computational bottleneck when scaling to large models with many adapted layers. Please refer to Appendix D for a detailed discussion.

**Complexity.** StelLA adds $r(m + n) + r^2$ parameters per adapted layer, which is $r^2$ parameters more than LoRA. This additional memory footprint is negligible since $r \ll \min(m, n)$. In our experiments, we show that the number of parameters in StelLA is comparable to LoRA and DoRA [40]. Nevertheless, in Section 5.5, we show that the superior performance of StelLA is not merely the result of this tiny increase in the number of trainable parameters but our geometric formulation by relaxing the orthonormality constraints. Regarding inference, similar to LoRA, we can merge the adapted weights into the original ones, resulting in a single weight matrix with no overhead.

**Gradient Scaling.** Previous work such as LoRA+ [24] showed that setting different learning rates for $B$ and $A$ in LoRA can improve convergence speed. The core idea is to ensure that both $B$ and $A$ are efficiently updated during training. Motivated by their insights, we balance the learning speed of $U$ and $V$ in StelLA via gradient scaling. For a random unit vector $x \in \mathbb{R}^m$, it is easy to show that the variance of each element is $\frac{1}{m}$. Since the columns of $U$ and $V$ are unit vectors, their individual entries are expected to have magnitudes on the order of $1/\sqrt{m}$ and $1/\sqrt{n}$, respectively. Adam-style optimizers, however, normalize gradients so that their coordinate-wise variance is $\Theta(1)$ [69]. This means that the learning speed—the ratio between the average magnitude of the effective gradient and the average magnitude of the parameter—for the entries of $U$ and $V$ are different when $m \neq n$. This imbalance is particularly pronounced in feed-forward layers of LLMs, where the hidden dimension is typically enlarged and then shrunk by a factor of 4 [9], causing the taller matrix to learn two times faster. To compensate for this difference, before applying the projection operation (line 7 in Algorithm 1), we scale the gradients of $U$ and $V$ by $\sqrt{d/m}$ and $\sqrt{d/n}$, respectively, where $d$ is a hyperparameter. In our experiments, we set $d$ equal to the hidden dimension of the input tokens.

**Comparison to Existing Geometric Optimizers.** Existing geometric optimization methods, such as Riemannian SGD [54] and Riemannian Adam [6], can handle optimization over generic manifolds. However, these methods are intrinsically coupled to specific optimizers, as they rely on accessing and manipulating internal states such as momentum or adaptive learning rates. For instance, Riemannian Adam parallel transports the momentum vector across tangent spaces to maintain consistency in updates. This tight integration limits their generalizability to other optimization algorithms. In contrast, our approach decouples geometric constraints from the choice of base Euclidean optimizer by treating its update direction as a perturbation to the Riemannian gradient. The projection operator $\pi$ (line 7 in Algorithm 1) ensures that the perturbed gradient is mapped back to the tangent space of the current point. This allows us to use any Euclidean optimizer without modifying its internal logic.

## 5 Experiments

We conduct extensive experiments to evaluate the performance of StelLA on various domains: natural language understanding (commonsense reasoning), natural language generation (math and code generation), visual understanding (image classification), and visual generation (text-to-image), using a diverse set of model architectures, including LLaMA2 [60], LLaMA3 [21], ViT [16], and Stable Diffusion [53]. We compare StelLA with a diverse set of low-rank adaptation baselines, covering various methodological categories: LoRA [29] as the standard baseline; DoRA [40] as a strong LoRA variant; PiSSA [43] and OLoRA [10] for SVD-based initialization; TriLoRA [20] and MoSLoRA [66] for three-factor decompositions; and ScaledAdamW [72] to represent geometry-aware optimization. Finally, we perform ablation studies to analyze the effect of different components in StelLA.

### 5.1 Commonsense Reasoning

**Models and Datasets.** We evaluate the performance of StelLA on the commonsense reasoning benchmark, which assesses the reasoning capabilities of large language models across 8 sub-tasks. Following the setup of Liu et al. [40], we train on the combined data from all sub-tasks and evaluate on the test set. We fine-tune two popular LLM checkpoints, LLaMA2-7B [60] and LLaMA3-8B [21].

Table 1: Accuracy on the commonsense reasoning benchmark. All results are averaged over 3 runs.

| Model | Method | Params (%) | BoolQ | PIQA | SIQA | HellaS. | WinoG. | ARC-e | ARC-c | OBQA | Avg. |
|-------|--------|-----------|-------|------|------|---------|--------|-------|-------|------|------|
| LLaMA2-7B | LoRA | 0.826 | 72.02 | 83.46 | 79.87 | 90.44 | 82.69 | 84.83 | 71.19 | 81.53 | 80.76 |
| | DoRA | 0.838 | 72.67 | 83.48 | 79.82 | 90.82 | 83.58 | 85.16 | 71.27 | 81.20 | 81.00 |
| | PiSSA | 0.826 | 71.16 | 83.89 | 79.19 | 91.00 | 82.87 | 85.09 | 69.48 | 83.93 | 80.83 |
| | OLoRA | 0.826 | 71.11 | 82.70 | 78.64 | 89.41 | 81.48 | 83.58 | 68.17 | 80.20 | 79.41 |
| | TriLoRA | 0.828 | 71.23 | 80.96 | 78.33 | 80.91 | 77.59 | 81.76 | 66.69 | 79.80 | 77.16 |
| | MoSLoRA | 0.828 | 71.54 | 83.84 | 79.60 | 90.50 | 83.19 | 84.40 | 69.96 | 80.47 | 80.44 |
| | ScaledAdamW | 0.826 | 72.20 | 83.86 | 79.67 | 90.80 | 82.43 | 85.55 | 70.59 | 81.93 | 80.88 |
| | **StelLA** | 0.828 | **73.62** | **84.87** | **80.64** | **91.44** | **84.50** | **86.43** | **72.84** | **84.33** | **82.33** |
| LLaMA3-8B | LoRA | 0.700 | 75.16 | 88.14 | 80.18 | 95.41 | 86.74 | 90.84 | 78.70 | 87.00 | 85.27 |
| | DoRA | 0.710 | 75.38 | 88.01 | 79.94 | 95.35 | 86.29 | 90.54 | 79.69 | 86.07 | 85.16 |
| | PiSSA | 0.700 | 74.67 | 88.12 | 80.50 | 94.98 | 85.22 | 90.15 | 78.87 | 85.60 | 84.76 |
| | OLoRA | 0.700 | 74.41 | 87.68 | 79.55 | 94.79 | 85.40 | 90.04 | 78.24 | 85.00 | 84.39 |
| | TriLoRA | 0.702 | 73.09 | 86.64 | 78.64 | 93.40 | 82.88 | 87.76 | 75.26 | 84.30 | 82.74 |
| | MoSLoRA | 0.702 | 74.88 | 88.43 | 80.31 | 95.50 | 86.26 | 90.00 | 79.86 | 85.80 | 85.13 |
| | ScaledAdamW | 0.700 | 75.24 | 88.57 | 80.21 | 95.81 | 85.11 | 91.09 | 80.55 | 86.60 | 85.40 |
| | **StelLA** | 0.702 | **75.91** | **89.86** | **81.68** | **96.41** | **87.82** | **91.98** | **82.34** | **87.80** | **86.72** |

**Implementation Details.** We compare StelLA against all the aforementioned baselines. For all methods, we insert low-rank adapters into the $Q$, $K$, and $V$ projections of the self-attention layers and the up and down projections of the feed-forward layers. For fair comparison, we fix the rank to 32, $\alpha$ to 64, batch size to 16, weight decay to 0, dropout to 0.05, and train for 3 epochs using AdamW. The learning rate is separately tuned for each method and follows a linear decay schedule.

**Results.** Table 1 shows that StelLA yields consistent, model-agnostic gains on every commonsense sub-task. StelLA lifts the average accuracy of LLaMA2-7B from 81.0% (the best baseline) to 82.3% and that of LLaMA3-8B from 85.4% to 86.7%, corresponding to absolute improvements of about 1.3 points on both models. Crucially, the benefit is uniform: StelLA attains the top score on all eight datasets for both model sizes, showing that its geometry-aware adapters generalize across binary, causal and multiple-choice commonsense formats.

## 5.2 Math and Code Generation

**Models and Datasets.** To assess the generative capabilities of StelLA, we conduct experiments on two representative natural language generation (NLG) tasks: mathematical reasoning and code generation. For math-related tasks, the models are fine-tuned on MetaMathQA [70] and evaluated on GSM8K [13] and MATH [26]. For code-related tasks, we train on CodeFeedback [75] and evaluate on HumanEval [11] and MBPP [3].

**Implementation Details.** We benchmark StelLA against three strong baselines: LoRA [29], DoRA [40], and PiSSA [43]. We follow the experimental protocol of Meng et al. [43]. Each method applies low-rank adaptation to all linear transformations in both self-attention and feed-forward modules. For consistency, we standardize the training configuration across methods: rank is set to 128, $\alpha$ to 128, LoRA dropout and weight decay are both zero, and models are trained with the AdamW optimizer for 1 epoch using a batch size of 128. The learning rate follows a cosine decay schedule and is tuned individually per method to ensure optimal performance.

**Results.** As summarized in Table 2, StelLA delivers the strongest overall performance (39.30 on average) across the math and code generation benchmarks, surpassing all three established baselines by a comfortable margin (up to +2.69 absolute points on average over DoRA). Importantly, it is the only method that ranks first or second on every benchmark, confirming StelLA's versatility and effectiveness in challenging natural language generation scenarios.

## 5.3 Image Classification

**Models and Datasets.** We assess the performance on image classification tasks using the Vision Transformer [16] pretrained on ImageNet-21K [14]. We fine-tune the Base and Large ViT on 8

Table 2: Results on math and code generation. All results are averaged over 3 runs.

| Model | Method | Params (%) | Math | | Code | | Avg. |
|---|---|---|---|---|---|---|---|
| | | | GSM8K | MATH | HumanEval | MBPP | |
| LLaMA2-7B | LoRA | 4.531 | 64.67 | 15.33 | 30.07 | _41.00_ | _37.76_ |
| | DoRA | 4.550 | _64.87_ | 15.32 | 27.00 | 39.27 | 36.61 |
| | PiSSA | 4.531 | 63.73 | **16.64** | _31.10_ | 38.20 | 37.41 |
| | **StelLA** | 4.581 | **65.43** | _16.55_ | **33.73** | **41.47** | **39.30** |

Table 3: Comparison of image classification accuracy (%) on 8 datasets averaged over 3 runs.

| Model | Method | Params (%) | DTD | EuroSAT | Flowers102 | Food101 | OxfordPets | Sun397 | CIFAR10 | CIFAR100 | Avg. |
|---|---|---|---|---|---|---|---|---|---|---|---|
| ViT Base | LoRA | 0.72 | _79.20_ | **98.98** | **99.22** | 90.07 | 92.94 | 75.62 | _98.92_ | 92.60 | 90.94 |
| | DoRA | 0.75 | 78.56 | 98.91 | _99.19_ | _90.15_ | _93.40_ | _75.67_ | **98.96** | **92.78** | _90.95_ |
| | PiSSA | 0.72 | 77.29 | _98.93_ | 98.88 | 89.99 | 93.21 | 75.41 | _98.92_ | 92.28 | 90.61 |
| | **StelLA** | 0.73 | **79.89** | 98.91 | **99.22** | **90.17** | **93.54** | **76.28** | 98.88 | _92.72_ | **91.20** |
| ViT Large | LoRA | 0.53 | 80.11 | _99.11_ | **99.32** | _91.30_ | **94.41** | 77.08 | _99.23_ | 94.06 | _91.83_ |
| | DoRA | 0.55 | _80.21_ | 99.00 | _99.28_ | 91.27 | 94.30 | _77.19_ | **99.29** | **94.11** | _91.83_ |
| | PiSSA | 0.53 | _80.21_ | _99.11_ | 99.20 | 91.10 | _94.33_ | 77.15 | 99.11 | 93.99 | 91.78 |
| | **StelLA** | 0.54 | **81.54** | **99.17** | 99.19 | **91.33** | 94.14 | **77.51** | 99.16 | 93.92 | **92.00** |

datasets: Caltech101 [19], CUB200 [62], Cars196 [31], Flowers102 [45], Food101 [8], Oxford-Pets [47], Sun397 [68], CIFAR10 and CIFAR100 [32], and measure the validation top-1 accuracy.

**Implementation Details.** We include the following baselines for comparison: LoRA [29], DoRA [40], and PiSSA [43]. We adapt only the query and value matrices of the attention layers in the ViT model. For all methods, we fix the rank and scaling factor $\alpha$ at 16, use a batch size of 128, apply no weight decay, set dropout to 0.1, and train for 10 epochs with the AdamW optimizer and a linear learning-rate schedule. The learning rate itself is tuned independently for each model to ensure a fair comparison.

**Results.** Table 3 summarizes the results of our experiments. We observe that StelLA delivers the best average performance using both ViT-Base and ViT-Large models, outperforming all other methods across most datasets. Using ViT-Base, StelLA attains the highest accuracy on five of the eight datasets and posts the best overall mean of 91.20%, surpassing the strongest baseline, DoRA, by +0.25 points. Scaling up to ViT-Large, StelLA achieves the best accuracy on four datasets and a mean of 92.00%, outperforming the best baseline, LoRA, by +0.17 points. These results demonstrate that StelLA is a strong contender for image classification tasks.

## 5.4 Text-to-Image Generation

**Models and Datasets.** To explore the effectiveness of StelLA in generative vision tasks, we fine-tune text-to-image models on five stylistically diverse datasets sourced from *CivitAI* [12]: *Barbie*, *Cyberpunk*, *Elementfire*, *Expedition*, and *Hornify*. Each dataset includes captions generated using the BLIP model [34]. We conduct experiments on two commonly used latent diffusion [53] based models—Stable Diffusion v1.5 and v2.0, which are equipped with a U-Net architecture composed of ResNet blocks [25].

**Implementation Details.** We follow standard LoRA fine-tuning recipes in *Diffusers* [61] by injecting LoRA parameters into the cross-attention layers of the U-Net. For benchmarking, we compare StelLA with LoRA, DoRA, and PiSSA and report CLIP score [52] and FID [27], two metrics that respectively measure semantic alignment and visual fidelity. For all methods, we fix the rank and scaling factor $\alpha$ at 4, use a batch size of 8, apply 0.01 weight decay and train for 100 epochs with the AdamW optimizer and a cosine learning-rate schedule. The learning rate itself is tuned independently for each model to ensure a fair comparison.

Table 4: Text-to-Image quantitative results on finetuning SD 1.5 and SD 2.0 to downstream tasks. In most cases, StelLA achieves the best FID and on-par CLIP scores.

| Model | Method | Params (%) | BarbieCore | | Cyberpunk | | ElementFire | | Expedition | | Hornify | |
|---|---|---|---|---|---|---|---|---|---|---|---|---|
| | | | FID ↓ | CLIP ↑ | FID ↓ | CLIP ↑ | FID ↓ | CLIP ↑ | FID ↓ | CLIP ↑ | FID ↓ | CLIP ↑ |
| | w/o finetune | - | 208.11 | **30.84** | 145.37 | 27.49 | 253.66 | 27.78 | 180.18 | **27.99** | 212.90 | **27.98** |
| | LoRA | 0.093 | 175.48 | 30.31 | 127.50 | 27.62 | 202.49 | 27.80 | 156.34 | 27.64 | 180.48 | 27.24 |
| SD 1.5 | DoRA | 0.104 | 175.04 | 30.36 | 127.11 | 27.61 | 200.77 | 27.78 | 155.80 | 27.65 | 179.58 | 27.26 |
| | PiSSA | 0.093 | 299.49 | 17.32 | 269.44 | 16.88 | 303.18 | 18.82 | 291.22 | 17.55 | 295.15 | 17.29 |
| | **StelLA** | 0.093 | **170.25** | 30.15 | **124.46** | **27.73** | **194.41** | **27.83** | **146.12** | 27.56 | **167.53** | 27.23 |
| | w/o finetune | - | 210.45 | 31.12 | 158.68 | 27.42 | 256.01 | 27.82 | 180.74 | **27.86** | 214.41 | **28.15** |
| | LoRA | 0.096 | **171.68** | 30.72 | 140.77 | 27.85 | **194.53** | 27.91 | 159.66 | 27.53 | 188.11 | 27.20 |
| SD 2.0 | DoRA | 0.107 | 176.16 | 30.87 | 140.71 | **27.94** | 197.12 | 27.92 | 159.54 | 27.54 | 185.40 | 27.19 |
| | PiSSA | 0.096 | 315.64 | 16.86 | 272.60 | 15.81 | 284.23 | 17.96 | 267.87 | 16.71 | 294.59 | 15.71 |
| | **StelLA** | 0.096 | 171.83 | 30.92 | **135.05** | 27.83 | 194.71 | **28.00** | 154.06 | 27.34 | **177.51** | 27.05 |

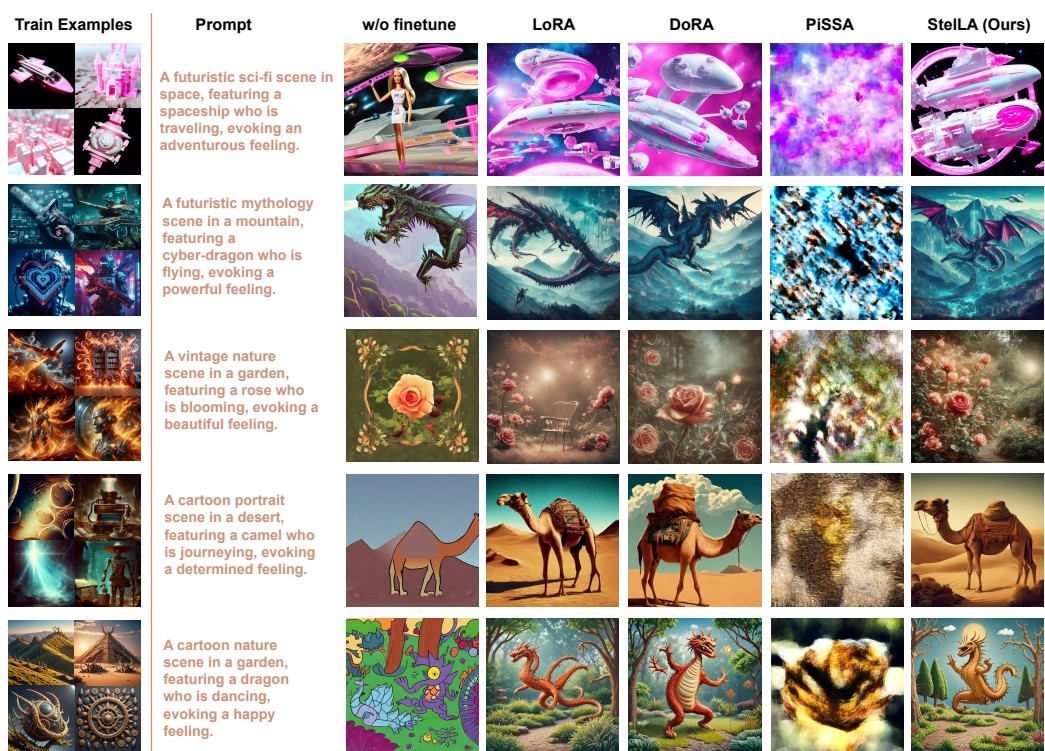

Figure 1: Qualitative results of finetuning SD 1.5 on *BarbieCore*, *Expedition* and *Hornify*.

**Results.** Quantitative results in Table 4 show that StelLA consistently yields the lowest FID scores, often outperforming the baselines by a notable margin—particularly on datasets like *Expedition* and *Hornify*—while maintaining competitive CLIP scores. In addition, qualitative results shown in Figure 1 reveal that StelLA generates images with strong stylistic consistency (*e.g.*, preserving the original background aesthetics in *BarbieCore*) and high perceptual quality, demonstrating its strength in adapting generative models to downstream domains.

## 5.5   Ablation Studies

In this section, we analyze the design choices in StelLA. We compare the performance of alternative geometric structures, different initialization strategies, and study the effectiveness of the additional gradient scaling for the Adam optimizer. We also evaluate the effect of parallel transport introduced in Riemannian Adam [6] and the effect of an alternative choice for the retraction operator. The

Table 5: Ablation study on design choices using the Commonsense with LLaMA3-8B. GS and PT denote gradient scaling and parallel transport, respectively. Results are averaged over 3 runs.

| Geometry | Initialization | GS | PT | BoolQ | PIQA | SIQA | HellaS. | WinoG. | ARC-e | ARC-c | OBQA | Avg. |
|---|---|---|---|---|---|---|---|---|---|---|---|---|
| StelLA | Non-zero | ✓ | × | 75.9 | **89.9** | 81.7 | 96.4 | **87.8** | 92.0 | **82.3** | 87.8 | **86.7** |
| Euclidean | Non-zero | N/A | × | 74.0 | 88.0 | 80.4 | 94.9 | 85.1 | 89.5 | 78.1 | 85.1 | 84.4 |
| Quotient | | | | 75.7 | 88.6 | 81.0 | 96.1 | 86.9 | 90.9 | 80.0 | 86.4 | 85.7 |
| StelLA | Zero | ✓ | × | 75.9 | 89.2 | 81.6 | 96.3 | 87.4 | 91.7 | 81.8 | 87.9 | 86.5 |
| | Pseudo-zero | | | 72.8 | 87.1 | 80.8 | 94.8 | 86.0 | 89.3 | 78.2 | 84.8 | 84.2 |
| | SVD-major | | | 76.1 | 89.8 | 81.5 | **96.5** | 87.5 | 92.3 | 81.9 | **88.3** | **86.7** |
| | SVD-minor | | | 75.9 | 89.8 | **81.9** | 96.4 | 87.2 | **92.4** | 82.0 | 87.5 | 86.6 |
| StelLA | Non-zero | × | × | 76.1 | 89.7 | 81.2 | 96.3 | 87.4 | 92.0 | 80.9 | 87.6 | 86.4 |
| StelLA | Non-zero | ✓ | ✓ | **76.2** | 89.6 | 81.4 | 96.4 | 87.4 | 91.9 | 81.7 | 87.1 | 86.5 |

Commonsense reasoning benchmark with the LLaMA3-8B model is used for all ablation studies, and the results are summarized in Table 5.

**Geometric Structures.** For each weight matrix $W$ to be adapted, there are three learnable matrices $U$, $S$, and $V$ in StelLA. Since $U$ and $V$ are constrained to lie on the Stiefel manifold, StelLA's geometry is a product of two Stiefel manifolds and a Euclidean manifold, namely, $\mathrm{St}(r, m) \times \mathbb{R}^{r \times r} \times \mathrm{St}(r, n)$. To show the effectiveness of this geometry, we compare it with the following alternatives:

- **Euclidean:** $\mathbb{R}^{m \times r} \times \mathbb{R}^{r \times r} \times \mathbb{R}^{n \times r}$. The simplest geometry is a product of Euclidean spaces, which does not impose any orthonormality constraints on the factors. It is used in previous works such as TriLoRA [20] and MoSLoRA [66].
- **Quotient:** $\mathrm{St}(r, m) \times \mathbb{R}^{r \times r} \times \mathrm{St}(r, n) / (\mathcal{O}(r) \times \mathcal{O}(r))$. The three-factor decomposition of a low-rank matrix $M = USV^\top$ is not unique due to the equivalence relationship $(U, S, V) \sim (UO_1, O_1^\top SO_2, VO_2), \forall O_1, O_2 \in \mathcal{O}(r)$, where $\mathcal{O}(r)$ is the orthogonal group. Factoring out this symmetry, we get a quotient space where each low-rank matrix is uniquely represented. We adapt StelLA to use this geometry along with the Riemannian metric defined in [44]. Details of the geometry and its optimization are discussed in Appendix B.

Comparing the performance of these geometries in Table 5, we observe that the product space $\mathrm{St}(r, m) \times \mathbb{R}^{r \times r} \times \mathrm{St}(r, n)$ for StelLA consistently outperforms the other two geometries across subtasks. This evidences that (1) The Euclidean three-factor geometry is not as effective as StelLA's geometry, implying that the orthonormality constraints on $U$ and $V$ are beneficial for the low-rank adaptation task, and (2) the StelLA's geometry is more effective than the quotient geometry, which is likely due to the fact that the Riemannian metric [44] on the quotient space was initially designed for the low-rank matrix completion problem rather than low-rank adaptation.

**Initializations.** We refer to the initialization of StelLA introduced in Section 4 as **non-zero** initialization, and compare it with the following initialization strategies:

- **Zero.** Initialize $S$ to be zero, and $U$ and $V$ to be random column-orthonormal matrices.
- **Pseudo-zero.** Same as the non-zero initialization, except that the original weight matrix is modified by subtracting the adapter's initialization.
- **SVD-major.** Initialize $U$ and $V$ to be the leading $r$ left and right singular vectors of the pretrained weight matrix, respectively. $S$ is initialized as the identity matrix. This setting aligns with the philosophy in PiSSA [43] and can be interpreted as StelLA + PiSSA.
- **SVD-minor.** Same as SVD-major, except that the trailing $r$ singular vectors are used. This setting aligns with the philosophy in MiLoRA [63] and can be interpreted as StelLA + MiLoRA.

From the results in Table 5, we observe that (1) zero initialization is not as effective as the non-zero initialization. We hypothesize the small value of $S$ creates small gradients for $U$ and $V$ at the beginning of training, leading to slow convergence. (2) Pseudo-zero initialization leads to the worst performance, presumably because it contaminates the pretrained weights with the adapter's

Table 6: Comparison of the polar retraction with the exponential map on the commonsense reasoning benchmark. Results are averaged over 3 runs.

| Geometry | Retraction | BoolQ | PIQA | SIQA | HellaS. | WinoG. | ARC-e | ARC-c | OBQA | Avg. |
|----------|-----------|-------|------|------|---------|--------|-------|-------|------|------|
| StelLA | Polar | 75.91 | 89.86 | 81.68 | 96.41 | 87.82 | 91.98 | 82.34 | 87.80 | 86.72 |
| StelLA | Exponential Map | 75.98 | 89.52 | 81.10 | 96.42 | 88.27 | 91.83 | 82.85 | 88.13 | 86.76 |

initialization. (3) SVD-major and SVD-minor initializations show similar performance as non-zero initialization. This showcases the robustness of our geometric optimization, as it can effectively learn the suitable subspaces regardless of the initialization.

**Gradient Scaling (GS) and Parallel Transport (PT).** As we have used the AdamW optimizer, we evaluate the effectiveness of the gradient scaling strategy introduced in Section 4 and the parallel transport introduced in Riemannian Adam [6]. Table 5 indicates that our gradient scaling slightly improves the average accuracy from $86.4\%$ to $86.7\%$, showing its effectiveness in balancing the learning rates of $U$ and $V$. Moreover, implementing the parallel transport results in the vanilla Riemannian Adam optimization. However, it does not lead to any performance gain compared to our treatment of converting a Euclidean Adam into a Riemannian one in Algorithm 1.

**Alternative Choices for Retraction.** Other than the polar retraction in Equation (4), the exponential map offers an alternative way of retraction. The exponential map is a locally defined operation that maps a tangent vector at one point to a new point on the manifold by following the geodesic in the given direction for a unit time. Intuitively, it traces the shortest curve on the manifold starting from a point $U$ along the direction of a tangent vector $\Delta$. Formally, let $QR := \Delta - UU^\top\Delta$ be the QR decomposition of $\Delta - UU^\top\Delta$. Then, the exponential map $\exp_U(\Delta)$ can be computed by $\exp_U(\Delta) = UM + QN$, where

$$\begin{pmatrix} M \\ N \end{pmatrix} := \exp \begin{pmatrix} U^\top\Delta & -R^\top \\ R & 0 \end{pmatrix} \begin{pmatrix} I \\ 0 \end{pmatrix}, \tag{7}$$

and $\exp$ is the matrix exponential (not to be confused with the exponential map $\exp_U$).

While the exponential map is geometrically well-founded, it is more expensive to compute than the polar retraction. We compare the performance of polar retraction and the exponential map in Table 6. The two methods achieve nearly identical results (86.72 vs. 86.76). We adopt the polar retraction as the default choice in StelLA due to its lower computational cost.

## 6 Limitation and Future Work

While StelLA demonstrates strong empirical performance across a range of models and tasks, it also comes with several limitations. First, we did not explore combining StelLA with complementary LoRA variants such as AdaLoRA [74]. These methods introduce orthogonal improvements such as rank scheduling. Second, our approach may be extended to tensor-valued layers using the low-rank Tucker format [71] straightforwardly, via the product Stiefel manifolds structure. Finally, due to time and resource constraints, we leave the evaluation of StelLA on more model families, and at very large scales, to future work.

## 7 Conclusion

We introduced StelLA, a subspace-aware extension of Low-Rank Adaptation (LoRA) that explicitly learns input and output subspaces through a three-factor decomposition $USV^\top$, with $U$ and $V$ constrained on Stiefel manifolds. By combining Riemannian optimization with a modular training algorithm that supports arbitrary optimizers, StelLA offers a flexible and geometry-aware approach to parameter-efficient fine-tuning. We demonstrated the effectiveness of StelLA on a variety of tasks, including language modeling, image classification, and text-to-image generation. Our experiments show that StelLA consistently outperforms LoRA and its variants, achieving state-of-the-art results on several benchmarks.

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

# A    Intuitive Example about Stiefel Manifold

To help readers understand the Stiefel manifold, we provide an intuitive example in low dimensions. In Section 3, we provided general equations for various geometric concepts. Here, we apply those ingredients to the following specific example.

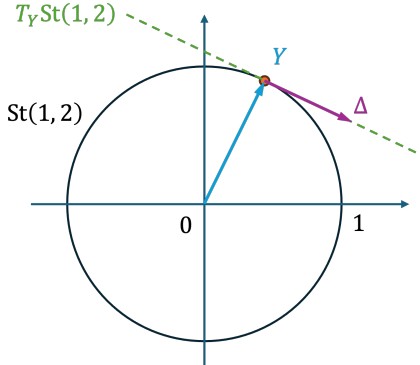

(a) The manifold $\mathrm{St}(1, 2)$ and the tangent space $T_Y\mathrm{St}(1, 2)$ at point $Y$.

(b) Converting a Euclidean gradient $\nabla_Y$ to the Riemannian gradient $\mathrm{grad}\, Y$ is a projection.

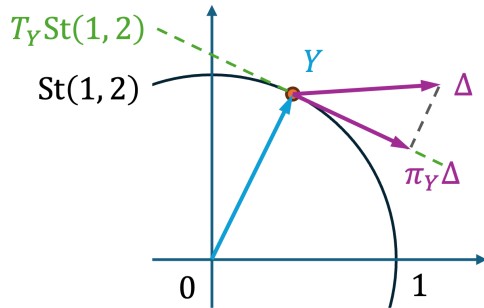

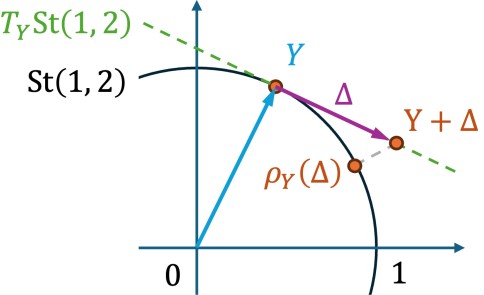

(c) Projecting a tangent vector $\Delta \in T_Y\mathbb{R}^2$ onto the tangent space of the manifold $T_Y\,\mathrm{St}(1, 2)$.

(d) Retraction $\rho_Y(\Delta)$ is the normalization of the vector $Y + \Delta$ to a unit vector.

Figure 2: An example Stiefel manifold $\mathrm{St}(1, 2)$: the unit circle on the two-dimensional plane.

**Manifold.**    Consider $\mathrm{St}(1, 2) = \{Y = [y_1, y_2]^\top \in \mathbb{R}^{2\times 1} | Y^\top Y = y_1^2 + y_2^2 = 1\}$, the Stiefel manifold of orthonormal 1-frames in $\mathbb{R}^2$, which consists of all unit vectors in the plane, *i.e.*, the unit circle. See Figure 2a for an illustration.

**Tangent Space.**    For a point $Y = [y_1, y_2]^\top \in \mathrm{St}(1, 2)$, a tangent vector $\Delta = [\delta_1, \delta_2]^\top$ must satisfy the constraint $Y^\top \Delta + \Delta^\top Y = 0$, which leads to the condition $y_1\delta_1 + y_2\delta_2 = 0$, namely $Y \perp \Delta$. The tangent space at $Y$ is the set of all vectors $\Delta$ that are orthogonal to $Y$. See Figure 2a for an illustration.

**Riemannian Metric.**    For two tangent vectors $\Delta_1, \Delta_2$ at a point $Y$, the canonical metric defines the inner product as

$$g_Y(\Delta_1, \Delta_2) = \mathrm{tr}(\Delta_1^\top (I_n - \frac{1}{2}YY^\top)\Delta_2) = \mathrm{tr}(\Delta_1^\top \Delta_2) - \frac{1}{2}\mathrm{tr}(\Delta_1^\top YY^\top \Delta_2) = \Delta_1^\top \Delta_2, \quad (8)$$

which equals to the standard inner product in $\mathbb{R}^2$. The last equation holds because $Y^\top \Delta_2 = 0$ and $\Delta_1^\top \Delta_2$ is a scalar.

**Riemannian Gradient.**    Suppose the Euclidean gradient of a function $f$ at $Y$ is $\nabla f(Y)$, which is abbreviated as $\nabla_Y$. Then the Riemannian gradient $\mathrm{grad}\, f(Y)$, abbreviated as $\mathrm{grad}\, Y$, is given by

$$\mathrm{grad}\, Y = \nabla_Y - Y\nabla_Y^\top Y = \nabla_Y - (\nabla_Y^\top Y)Y, \quad (9)$$

which is the projection of the Euclidean gradient onto the tangent space at $Y$. The second term $(\nabla_Y^\top Y)Y$ is the component of the Euclidean gradient that is not tangent to the manifold. See Figure 2b for an illustration.

**Tangent Space Projection.** The projection of a vector $\Delta$ onto the tangent space at $Y$ is given by

$$\pi_Y(\Delta) = \Delta - Y \operatorname{symm}(\Delta^\top Y) = \Delta - (\Delta^\top Y)Y, \tag{10}$$

which is the component of $\Delta$ that is tangent to the manifold. The second term $(\Delta^\top Y)Y$ is the component of $\Delta$ that is not tangent to the manifold. See Figure 2c for an illustration.

**Retraction.** For a tangent vector $\Delta$ at $Y$, the retraction is given by

$$\rho_Y(\Delta) = \operatorname{uf}(Y + \Delta) = \frac{Y + \Delta}{\|Y + \Delta\|}, \tag{11}$$

which in the $\mathrm{St}(1, 2)$ case is equivalent to normalizing the vector $Y + \Delta$ to a unit vector. See Figure 2d for an illustration.

## B Details of the Quotient Geometry

The three-factor decomposition of a low-rank matrix $M = USV^\top$ is not unique due to the equivalence relationship $(U, S, V) \sim (UO_1, O_1^\top SO_2, VO_2), \forall O_1, O_2 \in \mathcal{O}(r)$, where $\mathcal{O}(r)$ is the orthogonal group. Each rank-$r$ matrix has infinite representations. For example,

$$(UO_1)(O_1^\top SO_2)(VO_2)^\top = U(O_1O_1^\top)S(O_2O_2^\top)V^\top = USV^\top. \tag{12}$$

Factoring out this symmetry, we get a quotient space $\mathrm{St}(r, m) \times \mathbb{R}^{r \times r} \times \mathrm{St}(r, n)/(\mathcal{O}(r) \times \mathcal{O}(r))$, where each low-rank matrix is uniquely represented.

Mishra and Sepulchre [44] proposed a Riemannian metric for this quotient space. We adapt StelLA to use this Riemannian metric and refer to this setting as the Quotient geometry. The Riemannian metric in [44] is induced by the block approximation of the Hessian of $\|USV^\top - W\|_F^2$. For two tangent vectors $(\xi_U, \xi_S, \xi_V), (\eta_U, \eta_S, \eta_V)$ at a point $(U, S, V)$, the Riemannian metric is defined as

$$g_{(U,S,V)}\big((\xi_U, \xi_S, \xi_V), (\eta_U, \eta_S, \eta_V)\big) = \operatorname{tr}\left(SS^\top \xi_U^\top \eta_U\right) + \operatorname{tr} \xi_R^\top \eta_R + \operatorname{tr}\left(S^\top S \xi_V^\top \eta_V\right), \tag{13}$$

where $SS^\top$ and $S^\top S$ act as preconditioners which improve the convergence in the low-rank matrix completion problem. Details of the geometric optimization in this space is discussed in [44], including the equations for the Euclidean gradient to Riemannian gradient conversion, the tangent vector projection $\pi_{(U,S,V)}$, and the retraction $\rho_{(U,S,V)}$. Note that Mishra and Sepulchre [44] used the conjugate-gradient optimization algorithm to solve the low-rank matrix completion problem. However, with the operators such as $\pi_{(U,S,V)}$ and $\rho_{(U,S,V)}$ defined, it is straightforward to adapt to the Riemannian Adam algorithm following Algorithm 1. Specifically, we use [44, Eq. (9)] to convert the Euclidean gradient to the Riemannian gradient, [44, Eq. (5)] for the projection onto tangent space, and [44, Eq. (7)] for the retraction.

## C Details of Hyperparameters

We provide the detailed hyperparameters used in our experiments to ensure full reproducibility. For each benchmark, all compared methods share a common set of hyperparameters—such as rank, batch size, weight decay, and training schedule—which are outlined in Section 5. The only exception is the learning rate, which we individually tune for each method to ensure fair comparison. In this section, we list the specific learning rates used for each algorithm across benchmarks.

**Commonsense Reasoning.** We explore the learning rate in $\{0.0005, 0.0001, 0.00005, 0.00001\}$ for all algorithms, with the chosen values shown in Table 7.

**Math and Code Generation.** The search space for the learning rates is $\{0.0005, 0.0002, 0.00002\}$ for all methods. We list the selected learning rates in Table 8.

Table 7: Learning Rates for Commonsense Reasoning Experiments.

| Model | Method | Learning Rate |
|-------|--------|---------------|
| LLaMA2-7B | LoRA | 0.0001 |
| | DoRA | 0.0001 |
| | PiSSA | 0.00005 |
| | OLoRA | 0.00005 |
| | TriLoRA | 0.00001 |
| | MoSLoRA | 0.0001 |
| | ScaledAdamW | 0.0001 |
| | **StelLA** | 0.0005 |
| LLaMA3-8B | LoRA | 0.0001 |
| | DoRA | 0.0001 |
| | PiSSA | 0.00005 |
| | OLoRA | 0.00005 |
| | TriLoRA | 0.0001 |
| | MoSLoRA | 0.0001 |
| | ScaledAdamW | 0.00005 |
| | **StelLA** | 0.0005 |

Table 8: Learning Rates for Math and Code Generation Experiments.

| Model | Method | Math | Code |
|-------|--------|------|------|
| LLaMA2-7B | LoRA | 0.0002 | 0.0002 |
| | DoRA | 0.0002 | 0.0002 |
| | PiSSA | 0.0002 | 0.0002 |
| | **StelLA** | 0.0005 | 0.0005 |

**Image Classification.** The learning rate is tuned in $\{0.0001, 0.0005, 0.001, 0.005\}$ for all methods. The chosen learning rates are summarized in Table 9.

**Text to Image Generation.** We search the learning rate over $\{0.0001, 0.0002, 0.0004, 0.0008\}$ for all methods, with the selected values reported in Table 10. For each method, we use the same learning rate for all five CivitAI datasets.

## D  Details of the Computational Cost for Algorithm 1

In Algorithm 1, for each of the Stiefel parameter (*i.e.*, $U$ and $V$), it has the following operations:

1. Converting the Euclidean gradient to the Riemannian gradient using Equation (2).
2. Projecting the update direction onto the tangent space using Equation (3).
3. Retracting the updated point back onto the Stiefel manifold using Equation (4).

Among these, the first two steps involve only basic matrix multiplications and additions, and thus incur negligible computational overhead. The dominant cost arises from the retraction step, which requires a polar decomposition of the matrix $Y + \Delta$. This polar decomposition is typically computed via singular value decomposition (SVD) as follows.

Suppose the SVD of $(Y + \Delta) \in \mathbb{R}^{m \times r}$ with $m \gg r$ is expressed as $Y + \Delta = U\Sigma V^\top$, where $U \in \mathbb{R}^{m \times r}$ and $V \in \mathbb{R}^{r \times r}$ are orthogonal matrices, and $\Sigma \in \mathbb{R}^{r \times r}$ is a diagonal matrix. Then the orthonormal factor in the polar decomposition is given by

$$\mathrm{uf}(Y + \Delta) = UV^\top. \tag{14}$$

Thus, the retraction step has a cost dominated by computing the SVD of an $m \times r$ rectangular matrix.

Theoretically, the computational complexity of the polar retraction step is $O(mr^2)$ for tall matrices ($m \gg r$) due to the SVD computation. But practically, there are efficient SVD algorithms for tall matrices that can reduce the computational cost significantly. Specifically, we use the gesvda

Table 9: Learning Rates for Image Classification Experiments.

| Model | Method | DTD | EuroSAT | Flowers102 | Food101 | OxfordPets | Sun397 | CIFAR10 | CIFAR100 |
|---|---|---|---|---|---|---|---|---|---|
| ViT Base | LoRA | 0.005 | 0.005 | 0.005 | 0.001 | 0.005 | 0.001 | 0.001 | 0.001 |
| | DoRA | 0.005 | 0.005 | 0.005 | 0.001 | 0.005 | 0.001 | 0.001 | 0.001 |
| | PiSSA | 0.005 | 0.005 | 0.005 | 0.001 | 0.001 | 0.001 | 0.0005 | 0.001 |
| | **StelLA** | 0.005 | 0.005 | 0.005 | 0.001 | 0.005 | 0.001 | 0.001 | 0.001 |
| ViT Large | LoRA | 0.001 | 0.005 | 0.001 | 0.0005 | 0.001 | 0.0005 | 0.001 | 0.001 |
| | DoRA | 0.001 | 0.005 | 0.001 | 0.0005 | 0.0005 | 0.0005 | 0.001 | 0.0005 |
| | PiSSA | 0.001 | 0.001 | 0.001 | 0.0005 | 0.0005 | 0.0005 | 0.0001 | 0.0005 |
| | **StelLA** | 0.005 | 0.005 | 0.001 | 0.001 | 0.001 | 0.001 | 0.001 | 0.0005 |

Table 10: Learning Rates for Text to Image Generation Experiments.

| Model | Method | Learning Rate (same for all 5 tasks) |
|---|---|---|
| SD 1.5 & SD 2.0 | LoRA | 0.0001 |
| | DoRA | 0.0001 |
| | PiSSA | 0.0001 |
| | **StelLA** | 0.0008 |

solver [49], which is a CUDA-accelerated SVD implementation that can handle tall matrices efficiently. Below is a micro-benchmark of different SVD solvers for the matrix shapes used in StelLA for the commonsense benchmark on LLaMA3-8B. Results in Table 11 show that the `gesvda` solver is significantly faster than the default GPU solver in PyTorch.

The speed could be further improved by using batch processing to increase parallelism. Specifically, we can stack all the low-rank matrices $U$ and $V$ with the same shape into a batch, and then perform the polar retraction step on the batch of matrices. Table 12 is a micro-benchmark of the SVD using batch processing on the matrix shapes used in StelLA for the commonsense benchmark on LLaMA3-8B (there are 192, 64, and 64 matrices with shapes $4096 \times 32$, $1024 \times 32$, and $14336 \times 32$, respectively). The results show that the batched SVD is significantly faster than the sequential one, and the polar retraction step is no longer the bottleneck of the training time.

In practice, training a LoRA-adapted LLaMA3-8B model on a commonsense reasoning benchmark takes approximately 4.5 hours on a single H100 GPU, whereas training the same model with StelLA takes around 5.2 hours, about only 15% slower than vanilla LoRA.

## E  Discussion on Scale Stability

The scale stability of LoRA refers to the property that neither the activations nor the gradients explode or vanish as the rank $r$, input dimension $n$, and output dimension $m$ grow infinity [30, 64]. More precisely, *forward stability* is achieved if, assuming the input to the adapter has i.i.d. entries with second moment $\Theta_{r,m,n}(1)$, the output of the adapter also maintains a second moment of the same order. Similarly, *backward stability* is achieved if, when the gradient of the loss w.r.t. the adapter output has second moment $\Theta_{r,m,n}(1)$, the gradient w.r.t. the adapter input also remains at $\Theta_{r,m,n}(1)$. Letting $\gamma$ denote the scaling coefficient applied to the low-rank update, rsLoRA [30] shows that the original LoRA choice of $\gamma = \frac{\alpha}{r} = \Theta(\frac{1}{r})$ is not rank stable. Instead, they propose that $\gamma$ should scale as $\Theta(1/\sqrt{r})$ to maintain stability.

In StelLA, we initialize $U$ and $V$ as random orthogonal matrices and $S$ to the identity matrix. This setup satisfies the main assumptions in Theorem 3.2 of [64], which analyzes the scale stability of LoRA. Following their analysis, we assess the scale stability of StelLA at initialization as follows.

Let the adapter compute $y = \gamma U S V^{\top} x$ for an input vector $x$, where $\gamma$ is the scaling factor. Since $S$ is initialized to be identity, this simplifies to $y = \gamma U V^{\top} x$. The forward pass can be expressed component-wise as

$$y_i = \gamma \sum_{j=1}^{r} \sum_{k=1}^{n} U_{ij} V_{kj} x_k, \quad 1 \leq i \leq m. \quad \text{(Forward)} \tag{15}$$

Table 11: Runtime of different SVD solvers in PyTorch on H100 GPU.

| Matrix Shape | Default Solver (ms) | GESVDA Solver (ms) | Speedup of GESVDA |
|---|---|---|---|
| 4096×32 | 0.938 | 0.641 | 1.46× |
| 1024×32 | 0.795 | 0.632 | 1.26× |
| 14336×32 | 1.530 | 0.654 | 2.34× |

Table 12: Runtime of SVD with/without batch processing on H100 GPU. The GESVDA solver is used for both batched and sequential processing.

| Matrix Shape | Batch Size | Batched (ms) | Sequential (ms) | Speedup of Batched |
|---|---|---|---|---|
| 192×4096×32 | 192 | 5.80 | 123.1 | 22.22× |
| 64×1024×32 | 64 | 1.62 | 40.4 | 24.94× |
| 64×14336×32 | 64 | 2.89 | 41.8 | 14.46× |

To analyze the scale, we compute the second moment:

$$\mathbb{E}[y_i^2] = \gamma^2 \sum_{j_1=1}^{r} \sum_{j_2=1}^{r} \sum_{k_1=1}^{n} \sum_{k_2=1}^{n} \mathbb{E}[U_{ij_1} V_{k_1 j_1} x_{k_1} U_{ij_2} V_{k_2 j_2} x_{k_2}] \tag{16}$$

$$= \gamma^2 \sum_{j_1=1}^{r} \sum_{j_2=1}^{r} \sum_{k_1=1}^{n} \sum_{k_2=1}^{n} \mathbb{E}[U_{ij_1} U_{ij_2}]\mathbb{E}[V_{k_1 j_1} V_{k_2 j_2}]\mathbb{E}[x_{k_1} x_{k_2}] \tag{17}$$

$$= \gamma^2 \sum_{j_1=1}^{r} \sum_{j_2=1}^{r} \sum_{k=1}^{n} \mathbb{E}[U_{ij_1} U_{ij_2}]\mathbb{E}[V_{kj_1} V_{kj_2}]\mathbb{E}[x_k^2] \tag{18}$$

$$= \gamma^2 \sum_{j=1}^{r} \sum_{k=1}^{n} \mathbb{E}[U_{ij}^2]\mathbb{E}[V_{kj}^2]\mathbb{E}[x_k^2] \tag{19}$$

$$= \gamma^2 r n \frac{1}{m} \frac{1}{n} \Theta_{r,m,n}(1) = \frac{\gamma^2 r}{m} \Theta_{r,m,n}(1). \tag{20}$$

Hence, the forward pass remains scale-stable in the beginning of the training if $\gamma = \Theta(\sqrt{\frac{m}{r}})$. Equation (18) is derived from the independence of $x_{k_1}$ and $x_{k_2}$, where $\mathbb{E}[x_{k_1} x_{k_2}] = 0$ when $k_1 \neq k_2$. Equation (19) and Equation (20) are derived from the fact that for a random unit vector $a \in \mathbb{R}^n$, $\mathbb{E}[a_i a_j] = 0$ when $i \neq j$ and $\mathbb{E}[a_i^2] = \frac{1}{n}$.

For the backward pass, the gradient with respect to the input is given by $g = \frac{\partial \mathcal{L}}{\partial x} = \gamma V U^\top v$, where $v = \partial \mathcal{L}/\partial y$. Then,

$$g_i = \gamma \sum_{j=1}^{r} \sum_{k=1}^{m} V_{ij} U_{kj} v_k, \quad 1 \leq i \leq n. \quad \text{(Backward)} \tag{21}$$

and the second moment becomes:

$$\mathbb{E}[g_i^2] = \gamma^2 \sum_{j_1=1}^{r} \sum_{j_2=1}^{r} \sum_{k_1=1}^{m} \sum_{k_2=1}^{m} V_{ij_1} U_{k_1 j_1} v_{k_1} V_{ij_2} U_{k_2 j_2} v_{k_2} \tag{22}$$

$$= \gamma^2 \sum_{j=1}^{r} \sum_{k=1}^{m} \mathbb{E}[v_{ij}^2]\mathbb{E}[U_{kj}^2]\mathbb{E}[v_k^2]. \tag{23}$$

$$= \gamma^2 r m \frac{1}{n} \frac{1}{m} \Theta_{r,m,n}(1) = \frac{\gamma^2 r}{n} \Theta_{r,m,n}(1). \tag{24}$$

which implies the backward pass is stable in the beginning of the training if $\gamma = \Theta(\sqrt{\frac{n}{r}})$.

In practice, we adopt the coefficient $\gamma = \frac{\alpha}{r}$ to be consistent with LoRA. While this choice does not guarantee theoretical scale stability, it does not negatively affect training in practice, as $\alpha$ can be adjusted empirically. Since $m$, $n$, and $r$ are fixed for a given model, tuning $\alpha$ allows us to ensure stable and convergent training across tasks.

