# OpenReview forum: "StelLA: Subspace Learning in Low-rank Adaptation using Stiefel Manifold"
_NeurIPS.cc/2025/Conference — NeurIPS 2025 spotlight_

### Official Review · Reviewer_CC7z · 2025-06-20

**Clarity:** 3
**Significance:** 2
**Originality:** 2
**Rating:** 4
**Confidence:** 4

**Summary:**

The paper introduces Stella,  an low-rank optimization framework designed for factorizations of type $USV^T$ with orthonormal $U,V$.
The proposed optimizer performs a Riemannian optimization on the Stiefel manifolds for U,V and a standard gradient descent step for S.
The authors report superior performance than LoRA, which uses standard gradient descent for USV, which was the declared motivation to design the manifold constraint optimization.

**Questions:**

1. Although the authors give empirical evidence for improved evaluation metrics for a given rank budget, it is not clear to me why the manifold constraint should yield this. There is evidence and analyses that the plain LoRA does not preserve first-order optimality on the manifold of rank r matrices, see [3].
[3] Constructs a method that also preserves orthogonality of U,V -  as a consequence first order optimality conditions and convergence criteria on the rank r manifold is available.
 However in the proposed work, the authors consider the stiefel manifolds of U and V seperately, which gives a different view on optimality and convergence. This should be discussed.

[3] Schotthöfer et al., GeoLoRA: Geometric integration for parameter efficient fine-tuning, 2024.

2.In the gradient scaling section: Why is the "learning speed" an issue? Your method considers the training of U,S,V in some sense individually, i.e. each object is optimized in their own manifold/space. An example where USV=W is optimized as an element of a rank r manifold is given, e.g. in [3].

3. It would be interesting to compare the convergence plots of LoRA, the proposed method and full-rank training using the same optimizer (e.g. Adam) with the same learning rates and hyperparameters.
A simple test case could be compressing (not finetuning) the fully-connected layers (everything but key,query,value) of Vit-b/16 on Cifar100. This should train to convergence in about 500epochs, if you just use the pretrained imagenet weights, but converges slowly enough so that the difference between LoRa and your "proper" method should be visible. Finetuning of low-rank adapters may converge too fast to see the difference. Rank 50 should be a good initial guess.

4. There are methods e.g. AdaLoRa or [3] that are rank adaptive. Can your method handle adaptiveness?

5. Can your method be extended to tensor-valued layers, e.g. using low-rank Tucker format, see e.g. [4]

[4] Zangrando et al., Geometry-aware training of factorized layers in tensor Tucker format, 2024.

**Ethical Concerns:**

["NO or VERY MINOR ethics concerns only"]

**Final Justification:**

The authors have addressed my concerns and questions - and provided additional datapoints as requested.
Therefore I have raised my score to 4 (Borderline accept)

**Limitations:**

Some limitations are well discussed, e.g. the computational cost of the retraction. Others need to be added, see questions.

**Paper Formatting Concerns:**

no formatting concerns

**Quality:**

3

**Strengths And Weaknesses:**

The paper gives a good introduction to manifold constraint optimization and refers to relevant literature for further research.
The method is clearly structured and the implementation is well described.
Providing optimization methods that yield U,V from a Stiefel manifold is benefitial for certain downstream analyses, e.g. adversarial robustness [1,2] making this method relevant

[1]  Savostianova et al., Robust low-rank training via approximate orthonormal constraints, Advances in Neural Information Processing Systems 36, 66064–66083, 2023.
[2] Schotthöfer et al., Dynamical Low-Rank Compression of Neural Networks with Robustness under Adversarial Attacks, arXiv preprint arXiv:2505.08022, 2024.

(These can be included in the introduction, but I leave that to the discretion of the authors, as it is a downstream application)


Some questions and critiques can be found below:

---

> ### Author Rebuttal · Authors · 2025-07-31
>
> Thanks for the in-depth review and constructive comments.
> We have addressed all the comments and suggestions in the revised version of the manuscript.
> Below we clarify the questions and concerns in detail.
>
> > Optimization methods that yield U,V from a Stiefel manifold is beneficial for certain downstream analyses, e.g. adversarial robustness [1,2]. These can be included in the introduction.
>
> Thanks for the suggestion on potential applications.
> We will add these references to the introduction section in the revised version.
> Specifically, we will add the following sentences to L50:
>
> "Besides, maintaining orthogonality of the low-rank matrices $U$ and $V$ during training is also beneficial for certain downstream analyses, e.g. adversarial robustness [1,2]."
>
>
> > Although the authors give empirical evidence for improved evaluation metrics for a given rank budget, it is not clear to me why the manifold constraint should yield this. There is evidence and analyses that the plain LoRA does not preserve first-order optimality on the manifold of rank r matrices, see [3]. [3] Constructs a method that also preserves orthogonality of U,V - as a consequence first order optimality conditions and convergence criteria on the rank r manifold is available. However in the proposed work, the authors consider the Stiefel manifolds of U and V separately, which gives a different view on optimality and convergence. This should be discussed.
>
> Thanks for the pointer to the work of GeoLoRA [3].
> we will add GeoLoRA to the related work section and discuss the differences between GeoLoRA and StelLA.
>
> Regarding the convergence and optimality of StelLA, if we consider the tuple $(U, S, V)$ as a point on the product Riemannian manifold $\mathrm{St}(r, m) \times \mathbb{R}^{r \times r} \times \mathrm{St}(r, n)$, then the convergence can be analyzed using the Riemannian optimization theory, which is well-established in the literature.
> For example, we refer the readers to [a] and [b] for the convergence analysis of Riemannian SGD and Riemannian Adam optimization algorithms.
> Because of the product structure, the components $U$ and $V$ can be optimized separately, i.e., $U$ and $V$ are optimized on **their own respective Stiefel manifolds**, leading to our Alg. 1.
> This treatment is mathematically equivalent to optimizing on the product manifold directly, thus preserving the convergence properties.
> To avoid confusion, we would like to emphasize that all $U$s and $V$s in StelLA are optimized **jointly** in the same forward and backward pass, as described in Alg. 1.
>
> Furthermore, we would like to clarify that the product Riemannian manifold of StelLA is not the same as the manifold of rank-r matrices in GeoLoRA because each rank-r matrix has infinitely many representations in the product manifold due to the equivalence relationship by orthogonal groups (L287-289).
> Note that this does not affect the optimality because an optimal solution in the product manifold also corresponds to an optimal rank-r matrix.
>
> > In the gradient scaling section: Why is the "learning speed" an issue?
>
> Thanks for the question on the motivation of gradient scaling, we will clarify this detail in the main text of the revised version.
> Previous works such as LoRA+ [c] demonstrated that using the same learning rate for $A$ and $B$ matrices in LoRA does not allow efficient feature learning.
> They showed that setting **different learning rates** for $A$ and $B$ with a fixed ratio can improve performance.
> The core idea is to ensure that both $A$ and $B$ are efficiently updated.
> However, $U$ and $V$ in StelLA are constrained to be orthogonal matrices, while $A$ and $B$ in LoRA+ are unconstrained matrices, so the technique in LoRA+ cannot be directly applied to StelLA.
> Motivated by their insights, in StelLA, we ensure that both $U$ and $V$ are efficiently learned by balancing their effective learning speed via gradient scaling, achieving the same goal as LoRA+ but in a different way.
>
> > It would be interesting to compare the convergence plots of LoRA, the proposed method and full-rank training using the same optimizer (e.g. Adam) with the same learning rates and hyperparameters.
>
> Thanks for the suggestion on the comparison.
> We will add the convergence plots of StelLA, LoRA, and full-rank training in the appendix of the revised version.
>
> > A simple test case could be compressing (not finetuning) the fully-connected layers (everything but key,query,value) of Vit-b/16 on Cifar100. This should train to convergence in about 500epochs, if you just use the pretrained imagenet weights, but converges slowly enough so that the difference between LoRa and your "proper" method should be visible. Finetuning of low-rank adapters may converge too fast to see the difference. Rank 50 should be a good initial guess.
>
> Thanks for the suggestion.
> Unfortunately, we do not have experience in the experiment setting of compressing and didn't find reference code for benchmarking.
> Due to the limited time, we are unable to implement this experiment.
> We would like to kindly note that we have already conducted extensive experiments (and confirmed by Reviewer Qyxg) on the commonsense reasoning benchmark, math and code reasoning, image classification, and image generation benchmarks, which cover a wide range of tasks and models.
>
> > There are methods e.g. AdaLoRa or [3] that are rank adaptive. Can your method handle adaptiveness?
>
> Thanks for the question regarding rank adaptiveness.
> Yes, StelLA is compatible with AdaLoRA with minimal modifications.
> In AdaLoRA, the adapter is represented as $U\Lambda V^T$, where $U$ and $V$ are orthonormal matrices and $\Lambda$ is a diagonal matrix.
> Values in $\Lambda$ are used to control the rank allocation of the adapter.
> In StelLA, we can also constrain the matrix $S$ to be diagonal by setting all off-diagonal elements of $S$ to be zero during training.
> With this modification, StelLA can be used with AdaLoRA to adaptively adjust the rank of the adapter during training.
> However, it is hard to apply the rank adaptation approach in [3] to StelLA, because [3] needs expanding the rank of the adapter during training, which is not supported in StelLA (there is no rank augmentation step in StelLA).
>
> > Can your method be extended to tensor-valued layers, e.g. using low-rank Tucker format, see e.g. [4]
>
> Thanks for the question regarding the extension of StelLA to tensor-valued layers.
> Yes, StelLA can be extended to tensor-valued layers straightforwardly as follows.
> The Tucker decomposition of a rank-$(r_1,\dots r_d)$ tensor is written as $C\times_1 U_1\cdots \times_d U_d$, where $U_i$ are column-orthonormal matrices.
> Therefore, the tuple $(C, U_1, \dots, U_d)$ lies in the product Riemannian manifold of $\mathbb{R}^{r_1\times\cdots r_d}\times \mathrm{St}(r_1, n_1)\times \cdots \times \mathrm{St}(r_d, n_d)$.
> We can use the same approach as in StelLA to optimize the low-rank Tucker decomposition, because each component $U_i$ can be optimized in its own Stiefel manifold thanks to the product manifold structure.
>
> - [a] Bonnabel S. Stochastic gradient descent on Riemannian manifolds. IEEE Transactions on Automatic Control, 2013.
> - [b] Becigneul G, Ganea O E. Riemannian Adaptive Optimization Methods. ICLR 2019.
> - [c] Hayou S, Ghosh N, Yu B. Lora+: Efficient low rank adaptation of large models. ICML 2024.

---

> > ### Comment · Reviewer_CC7z · 2025-08-03
> >
> > Thank you for your answers.
> >
> > >Thanks for the suggestion on the comparison. We will add the convergence plots of StelLA, LoRA, and full-rank training in the appendix of the revised version.
> >
> > It would be beneficial for the review process to provide some data before the discussion period is over.
> > Since no links to outside websites are allowed, can the authors paste the data series of Stella, LoRA and full rank training/ft convergence for one or more of their experiments in csv formatting here? I can plot it for myself offline.

---

> > > ### Author Response · Authors · 2025-08-03
> > > **Convergence Plot Data**
> > >
> > > Thanks for the follow-up question. Below is the data for the convergence plot of full fine-tuning, LoRA, and StelLA in CSV format. They are trained using the same optimizer (AdamW-based Alg. 1 for StelLA and AdamW for the other two) with the same learning rates (0.0005) and hyperparameters on the commonsense reasoning benchmark. The first column represents the training step, and the values in the rest of the columns are the evaluation losses on the validation set consisting of 2000 samples.
> > >
> > > ```csv
> > > Step,Full,LoRA,StelLA
> > > 200,1.712,1.078,1.143
> > > 400,2.028,1.086,1.128
> > > 600,2.078,1.103,1.13
> > > 800,2.008,1.117,1.114
> > > 1000,1.932,1.114,1.112
> > > 1200,1.906,1.121,1.107
> > > 1400,1.913,1.131,1.099
> > > 1600,1.858,1.138,1.095
> > > 1800,1.823,1.131,1.097
> > > 2000,1.8,1.132,1.092
> > > 2200,1.782,1.13,1.086
> > > 2400,1.777,1.127,1.082
> > > 2600,1.767,1.131,1.075
> > > 2800,1.758,1.136,1.076
> > > 3000,1.732,1.137,1.075
> > > 3200,1.727,1.142,1.067
> > > 3400,1.722,1.14,1.067
> > > 3600,1.71,1.146,1.063
> > > 3800,1.706,1.145,1.062
> > > 4000,1.686,1.147,1.062
> > > 4200,1.683,1.132,1.055
> > > 4400,1.686,1.131,1.057
> > > 4600,1.683,1.131,1.052
> > > 4800,1.671,1.135,1.054
> > > 5000,1.655,1.12,1.058
> > > 5200,1.637,1.121,1.053
> > > 5400,1.65,1.123,1.046
> > > 5600,1.637,1.124,1.047
> > > 5800,1.619,1.125,1.046
> > > 6000,1.634,1.123,1.042
> > > 6200,1.604,1.13,1.044
> > > 6400,1.601,1.128,1.038
> > > 6600,1.597,1.133,1.037
> > > 6800,1.586,1.127,1.032
> > > 7000,1.593,1.137,1.034
> > > 7200,1.581,1.137,1.034
> > > 7400,1.587,1.134,1.035
> > > 7600,1.598,1.137,1.032
> > > 7800,1.577,1.128,1.027
> > > 8000,1.559,1.123,1.03
> > > 8200,1.558,1.124,1.03
> > > 8400,1.56,1.122,1.026
> > > 8600,1.553,1.112,1.021
> > > 8800,1.55,1.104,1.029
> > > 9000,1.527,1.114,1.024
> > > 9200,1.537,1.117,1.023
> > > 9400,1.524,1.116,1.023
> > > 9600,1.514,1.109,1.025
> > > 9800,1.5,1.108,1.022
> > > 10000,1.49,1.11,1.024
> > > 10200,1.499,1.108,1.01
> > > 10400,1.498,1.101,1.014
> > > 10600,1.484,1.104,1.011
> > > 10800,1.477,1.1,1.015
> > > 11000,1.483,1.114,1.013
> > > 11200,1.481,1.109,1.01
> > > 11400,1.465,1.102,1.01
> > > 11600,1.467,1.093,1.017
> > > 11800,1.476,1.082,1.009
> > > 12000,1.47,1.08,1.011
> > > 12200,1.459,1.088,1.01
> > > 12400,1.462,1.083,1.005
> > > 12600,1.455,1.084,1.008
> > > 12800,1.452,1.077,1.009
> > > 13000,1.45,1.07,1.006
> > > 13200,1.436,1.076,1.007
> > > 13400,1.422,1.063,1.003
> > > 13600,1.431,1.072,0.999
> > > 13800,1.42,1.077,0.999
> > > 14000,1.414,1.074,1.002
> > > 14200,1.41,1.067,1.0
> > > 14400,1.401,1.065,1.0
> > > 14600,1.411,1.063,1.001
> > > 14800,1.403,1.061,1.0
> > > 15000,1.4,1.057,1.002
> > > 15200,1.397,1.061,0.998
> > > 15400,1.393,1.057,0.993
> > > 15600,1.398,1.052,0.989
> > > 15800,1.381,1.054,0.992
> > > 16000,1.367,1.046,0.993
> > > 16200,1.365,1.04,0.989
> > > 16400,1.364,1.037,0.983
> > > 16600,1.356,1.042,0.982
> > > 16800,1.354,1.045,0.981
> > > 17000,1.354,1.038,0.979
> > > 17200,1.351,1.045,0.981
> > > 17400,1.345,1.034,0.979
> > > 17600,1.338,1.027,0.976
> > > 17800,1.337,1.029,0.978
> > > 18000,1.331,1.025,0.975
> > > 18200,1.32,1.019,0.972
> > > 18400,1.309,1.024,0.97
> > > 18600,1.311,1.013,0.973
> > > 18800,1.308,1.021,0.973
> > > 19000,1.305,1.012,0.97
> > > 19200,1.301,1.007,0.968
> > > 19400,1.296,0.997,0.969
> > > 19600,1.29,1.001,0.971
> > > 19800,1.287,1.0,0.966
> > > 20000,1.284,0.998,0.966
> > > 20200,1.278,0.999,0.959
> > > 20400,1.273,0.999,0.957
> > > 20600,1.274,0.992,0.959
> > > 20800,1.273,0.994,0.945
> > > 21000,1.27,0.98,0.947
> > > 21200,1.267,0.981,0.941
> > > 21400,1.281,0.995,0.946
> > > 21600,1.278,0.992,0.938
> > > 21800,1.284,0.983,0.935
> > > 22000,1.283,0.986,0.935
> > > 22200,1.273,0.983,0.935
> > > 22400,1.272,0.976,0.935
> > > 22600,1.267,0.972,0.933
> > > 22800,1.268,0.97,0.931
> > > 23000,1.268,0.972,0.93
> > > 23200,1.259,0.971,0.932
> > > 23400,1.256,0.969,0.934
> > > 23600,1.257,0.965,0.93
> > > 23800,1.248,0.969,0.928
> > > 24000,1.244,0.966,0.931
> > > 24200,1.233,0.956,0.931
> > > 24400,1.238,0.954,0.928
> > > 24600,1.232,0.961,0.927
> > > 24800,1.228,0.958,0.924
> > > 25000,1.227,0.955,0.922
> > > 25200,1.225,0.952,0.923
> > > 25400,1.222,0.949,0.919
> > > 25600,1.218,0.944,0.921
> > > 25800,1.22,0.945,0.921
> > > 26000,1.221,0.944,0.916
> > > 26200,1.213,0.944,0.916
> > > 26400,1.209,0.946,0.913
> > > 26600,1.209,0.943,0.911
> > > 26800,1.205,0.938,0.912
> > > 27000,1.204,0.937,0.907
> > > 27200,1.206,0.938,0.908
> > > 27400,1.204,0.94,0.909
> > > 27600,1.199,0.935,0.908
> > > 27800,1.2,0.927,0.905
> > > 28000,1.197,0.925,0.904
> > > 28200,1.197,0.925,0.904
> > > 28400,1.198,0.923,0.904
> > > 28600,1.196,0.923,0.9
> > > 28800,1.197,0.923,0.895
> > > 29000,1.196,0.918,0.894
> > > 29200,1.193,0.915,0.892
> > > 29400,1.193,0.914,0.889
> > > 29600,1.193,0.915,0.886
> > > 29800,1.193,0.914,0.884
> > > 30000,1.192,0.911,0.884
> > > 30200,1.192,0.912,0.883
> > > 30400,1.193,0.912,0.882
> > > 30600,1.192,0.91,0.882
> > > 30800,1.191,0.908,0.882
> > > 31000,1.191,0.908,0.882
> > > 31200,1.19,0.907,0.88
> > > 31400,1.19,0.906,0.88
> > > 31600,1.19,0.905,0.879
> > > 31800,1.19,0.905,0.879
> > > ```

---

> > > > ### Comment · Reviewer_CC7z · 2025-08-05
> > > >
> > > > Thank you for the additional information.
> > > >
> > > > I'm willing to raise my score.

---

### Official Review · Reviewer_eSmE · 2025-07-02

**Clarity:** 3
**Significance:** 3
**Originality:** 2
**Rating:** 5
**Confidence:** 4

**Summary:**

The work proposes a new LoRA method for fine-tuning large language models. It
		consists of exploiting three matrices $U \in \mathbb{R}^{m \times r},
		V \\in \\mathbb{R}^{n \\times r}, S \\in \\mathbb{R}^{r \\times r}$ in the following minimization
		task
		\begin{equation*}
			\\mathcal{L}\\left( W + USV^\\top \\right)	\\to
			\min\limits_{U \\in \\mathrm{St}\\, (r, m), \\; S, \\; V \\in \\mathrm{St}\\, (r, n)},
		\\end{equation*}
		where $W \\in \\mathbb{R}^{m \\times n}$ is a frozen weight matrix and
		$\\mathrm{St}\\, (r, m)$ is the Stiefel manifold. The authors
		present a new method (Algorithm $1$). They compute
		\\begin{equation*}
				\\hat{U}\_{k + 1} \\leftarrow \\mathrm{Optimizer}\\,(U_k, \\mathrm{grad}\\, U_k),
				\\hat{V}\_{k + 1} \\leftarrow \\mathrm{Optimizer}\\,(V_k, \\mathrm{grad}\\, V_k),
				\\hat{S}\_{k + 1} \\leftarrow \\mathrm{Optimizer}\\,(S_k, \\mathrm{grad}\\, S_k).
		\\end{equation*}
		The directions $\\hat{U}\_{k + 1}, \\hat{V}\_{k + 1}$ do not
		necesserally belong to the manifold
		(since the optimizer does not guarantee this). So the next step is to obtain a Riemannian
		direction via the projection onto a tangent space
		$\\mathrm{proj}\_{U_k} (U_k - \\hat{U}\_{k + 1})$ and $\\mathrm{proj}\_{V\_k} (V\_k - \\hat{V}\_{k + 1})$.
		Once the direction is found, the following steps correspond to vanilla Riemannian optimization techniques.

**Questions:**

- The mentioned quotient structure allows for decreasing the ambiguity of the
			choice for $U$ and $V$ matrices, but according to the text such an approach
			"is not designed for low-rank adaptation". Could you clarify the details for the optimization method that you use in this case?
- Did you try to use basic Riemannian optimization algorithms for
			$\\mathrm{St}\\, (r, m) \\times \\mathbb{R}^{r \\times r} \\times \\mathrm{St}\\, (r, n)$ and
			$(\\mathrm{St}\\, (r, m) \\times \\mathbb{R}^{r \\times r} \\times \\mathrm{St}\\, (r, n)) /(\\mathcal{O}(r)
			\\times \\mathcal{O}(r))$ without mixing them with the Euclidean optimizers?

**Ethical Concerns:**

["NO or VERY MINOR ethics concerns only"]

**Final Justification:**

The authors addressed my questions and concerns. The idea seems to be interesting and quite generally applicable.

**Limitations:**

Yes

**Quality:**

3

**Strengths And Weaknesses:**

Strengths
- The proposed method allows for using existing Euclidean optimizers, which
				is natively connected to the HuggingFace's PEFT library.
- The proposed method imposes stability due to the Stiefel manifold.


Weaknesses
- A single initialization strategy is used (no combinations with Pissa, etc). This might additionally boost the performance.
- The method implies using additional computation overheads in the retraction process (which is common for all Riemannian approaches).

---

> ### Author Rebuttal · Authors · 2025-07-31
>
> We would like to thank the reviewer for the constructive comments and suggestions.
> We have addressed all the comments and suggestions in the revised version of the manuscript.
> Below we clarify the questions and concerns in detail.
>
> > A single initialization strategy is used (no combinations with Pissa, etc). This might additionally boost the performance.
>
> Thanks for the suggestion.
> We indeed have experimented with different initialization strategies, including the combination of StelLA with PiSSA and MiLoRA in the ablation studies.
> In Table 5 of the main paper, the row marked as "SVD-major" is in fact StelLA + PiSSA, and the row marked as "SVD-Minor" is StelLA + MiLoRA.
> We will clarify this matter in the revised version to avoid confusion.
>
> For easier reference, we summarize the results in the table below.
>
> **Table: Comparison different initializations on the commonsense benchmark using LLaMA3-8B.**
>
> | Initialization | Accuracy (Avg.) | Comments on Initialization |
> | -------------- | --------------- | -------------------------- |
> | SVD-major      | 86.7            | StelLA + PiSSA             |
> | SVD-minor      | 86.6            | StelLA + MiLoRA            |
> | Non-zero       | 86.7            | StelLA                     |
>
> The results show that StelLA is robust to different initialization strategies.
> The subspace learning powered by Riemannian optimization can effectively learn the optimal subspace regardless of the initialization strategy (L312-313).
>
> > The method implies using additional computation overheads in the retraction process (which is common for all Riemannian approaches).
>
> Thanks for raising the concern.
> The retraction step is indeed computationally expensive, but with speed optimization including both algorithmic and hardware acceleration techniques, we can mitigate this issue and let StelLA be trained with only 15% additional training time compared to LoRA.
> Please kindly refer to the appendix Sec. D and our response to Reviewer Qyxg for the detailed and thorough discussion of computational cost of StelLA.
>
> > The mentioned quotient structure allows for decreasing the ambiguity of the choice for U and V matrices, but according to the text such an approach "is not designed for low-rank adaptation". Could you clarify the details for the optimization method that you use in this case?
>
> Thanks for the question, we will clarify this detail in the main text of the revised version.
> We use Alg. 1 to optimize the quotient structure, same as in StelLA.
> The difference between the quotient structure and StelLA is that the quotient structure is equipped with the Riemannian metric proposed in "R3MC: A Riemannian three-factor algorithm for low-rank matrix completion" [36].
> This Riemannian metric is designed for the low-rank **matrix completion** problem, which is different from the **low-rank adaptation** (LoRA) problem.
> We provided the details of R3MC's Riemannian metric in the appendix Sec. B.
> Specifically, as noted in the appendix L34-39, the Riemannian metric is induced by the block approximation of the Hessian of the loss function in the low-rank matrix completion problem: $||USV^T-W||_F^2$.
> Hence, we say that it is not designed for low-rank adaptation, because the Hessian of the loss function in low-rank adaptation is different from that in low-rank matrix completion.
>
> Nevertheless, the quotient structure can be optimized using Alg. 1, as long as the three equations for
> - converting Euclidean gradients to Riemannian gradients (see [36, Eq. 9]),
> - projecting a vector onto the tangent space (see [36, Eq. 5]), and
> - retracting a point onto the manifold (see [36, Eq. 7]),
>
> are computed accordingly for R3MC's Riemannian metric.
>
> > Did you try to use basic Riemannian optimization algorithms for $\mathrm{St}\, (r, m) \times \mathbb{R}^{r \times r} \times \mathrm{St}\, (r, n)$ and $(\mathrm{St}\, (r, m) \times \mathbb{R}^{r \times r} \times \mathrm{St}\, (r, n)) /(\mathcal{O}(r) \times \mathcal{O}(r))$ without mixing them with the Euclidean optimizers?
>
> Yes, we have tried using the Riemannian Adam optimizer [6] on StelLA in the ablation studies.
> The results are shown in the last row of Table 5, where the parallel transport is enabled.
> The setting of StelLA + parallel transport is equivalent to using the basic Riemannian Adam optimizer on the product manifold $\mathrm{St}\, (r, m) \times \mathbb{R}^{r \times r} \times \mathrm{St}\, (r, n)$.
> We will make this information explicit in the discussion of Table 5 in the revised manuscript.

---

> > ### Comment · Reviewer_eSmE · 2025-08-06
> >
> > Thank you for your rebuttal. It covered my questions and concerns. I think that the idea is interesting and quite generally applicable. Therefore, I would like to increase my score.

---

### Official Review · Reviewer_Bh7E · 2025-07-04

**Clarity:** 3
**Significance:** 3
**Originality:** 3
**Rating:** 4
**Confidence:** 3

**Summary:**

This paper enhances LoRA by applying an SVD-inspired three-factor formulation. Unlike prior methods that only use SVD for initialization, this approach keeps U and V orthonormal throughout training by constraining them to lie on the Stiefel manifold. Empirical results show the proposed method improves performance across a range of language and vision tasks.

**Questions:**

See Weaknesses

**Ethical Concerns:**

["NO or VERY MINOR ethics concerns only"]

**Final Justification:**

Thank authors for their reply. I will maintain my score.

**Limitations:**

Yes

**Quality:**

3

**Strengths And Weaknesses:**

Strengths:

The core novelty lies in applying LoRA decomposition $USV^T$, where the input/output subspaces U and V are constrained to lie on the Stiefel manifold. This allows the model to maintain orthonormality during training, unlike prior methods such as TriLoRA [1] and MoSLoRA [2], which do not keep the orthogonality during training.

The proposed method consistently outperforms other approaches on commonsense reasoning tasks, demonstrating its effectiveness in language understanding tasks.

The paper is well-written and easy to follow.

Weaknesses:

1) The paper lacks a comparison of computational efficiency. While the proposed method introduces additional computation for manifold operations (e.g., projection and retraction), which may impact training speed. A runtime or efficiency analysis would strengthen the empirical evaluation.

2) The method is related to LoRA-XS [3]. However, the paper does not include experimental comparisons against LoRA-XS, weakening the completeness of the evaluation.

3) It would be beneficial to compare the orthogonality of the proposed method to other methods like TriLoRA and LoRA-XS, as this would strengthen the effectiveness of the proposed Stiefel manifold constraints.

4) The performance improvements on image classification benchmarks are limited. This raises questions about the robustness and generalizability of the proposed method

[1] Trilora: Integrating svd for advanced style personalization in text-to-image generation

[2] Mixture-of-subspaces in low-rank adaptation.

[3] LoRA-XS: Low-Rank Adaptation with Extremely Small Number of Parameters

---

> ### Author Rebuttal · Authors · 2025-07-31
>
> Thanks for the insightful review and constructive comments.
> We have addressed all the comments and suggestions in the revised version of the manuscript.
> Below we clarify the questions and concerns in detail.
>
> > The paper lacks a comparison of computational efficiency. While the proposed method introduces additional computation for manifold operations (e.g., projection and retraction), which may impact training speed. A runtime or efficiency analysis would strengthen the empirical evaluation.
>
> Thanks for the suggestion.
> A detailed discussion of computational cost of Alg. 1 is provided in the appendix Sec. D.
> We will move this information from the appendix to the main text for better visibility.
>
> Here we summarize the key information:
> - The main computational overhead of Alg. 1 comes from the SVD operations in the polar retraction step, which can be reduced by using efficient CUDA-accelerated SVD implementation.
> - Training the LLaMA3-8B model with StelLA using Alg. 1 takes 6.5h for the commonsense benchmark, which is approximately 1.4x the time of LoRA (4.5h) on one H100 GPU. It can be further improved, e.g., by increasing parallelism with batch processing (details in our response to Reviewer Qyxg), which reduces the training time from 6.5h to **5.2h**, about only 15% slower than vanilla LoRA.
>
> Please kindly refer to our response to Reviewer Qyxg for more details on the computational cost, training time and how to improve them.
>
> > The method is related to LoRA-XS [3]. However, the paper does not include experimental comparisons against LoRA-XS, weakening the completeness of the evaluation.
>
> Thanks for the suggestion.
> Below we list the results on commonsense reasoning benchmark using LLaMA3-8B model from LoRA-XS's paper (they followed the same experiment setting as ours, so the results are directly comparable.)
> The result shows that StelLA outperforms LoRA-XS by a large margin.
> This is understandable because the number of learnable parameters in LoRA-XS is much smaller (only the $S$ matrix is learnable) than that in StelLA.
>
> **Table: Comparison of StelLA and LoRA-XS on the commonsense reasoning benchmark using LLaMA3-8B model.**
>
> | Method  | BoolQ | PIQA | SIQA | HellaS. | WinoG. | ARC-e | ARC-c | OBQA | Avg. |
> | ------- | ----- | ---- | ---- | ------- | ------ | ----- | ----- | ---- | ---- |
> | LoRA-XS | 66.6  | 85.8 | 79.4 | 90.1    | 85.2   | 87.0  | 76.5  | 81.8 | 81.6 |
> | StelLA  | 75.9  | 89.9 | 81.7 | 96.4    | 87.8   | 92.0  | 82.3  | 87.8 | 86.7 |
>
> > It would be beneficial to compare the orthogonality of the proposed method to other methods like TriLoRA and LoRA-XS, as this would strengthen the effectiveness of the proposed Stiefel manifold constraints.
>
> Thanks for the suggestion.
> We have briefly discussed the difference between StelLA and TriLoRA/LoRA-XS in the related work section (L94-97).
> Here we reiterate the key points and will add more details in the revised version:
> - TriLoRA does not initialize the low-rank matrices with orthogonal matrices (they initialize U with Gaussian, and V with zeros), and it does not maintain orthogonality during training.
> - LoRA-XS initializes the low-rank matrices with the leading $r$ singular vectors and the singular values of the weight matrix (Suppose $M=U_r S_r V_r^T$, then $U:=U_r$ and $V=S_r V_r^T$), in which only one of the low-rank matrices $U$ is orthogonal.
> Both $U$ and $V$ are frozen during training, which limits the expressiveness of the low-rank adaptation.
>
> We summarize the usage of orthogonality in StelLA, TriLoRA, and LoRA-XS in the table below.
>
> **Table: Comparison of orthogonality in StelLA, TriLoRA, and LoRA-XS**
>
> | Method  | Orthogonality of $U$ | Orthogonality of $V$ | Maintains Orthogonality in Training | $U, V$ Learnable | Commonsense Accuracy (avg.) |
> | ------- | -------------------- | -------------------- | ----------------------------------- | ---------------- | --------------------------- |
> | TriLoRA | $\times$             | $\times$             | $\times$                            | $\checkmark$     | 82.7                        |
> | LoRA-XS | $\checkmark$         | $\times$             | $\checkmark$ (by freezing $U$)      | $\times$         | 81.6                        |
> | StelLA  | $\checkmark$         | $\checkmark$         | $\checkmark$                        | $\checkmark$     | 86.7                        |
>
> > The performance improvements on image classification benchmarks are limited. This raises questions about the robustness and generalizability of the proposed method
>
> Thanks for raising the concern.
> The relatively limited performance improvements on image classification benchmarks are due to the fact that the performance might have being saturated on the benchmark: the accuracy of LoRA is above 90% on 6/8 of the datasets.
> Yet, the performance improvements of StelLA over LoRA are significant on the two less saturated datasets: DTD (79.2% -> 79.89%) and Sun397 (75.62% -> 76.28%).
> Despite facing the challenge in further improving performance, our method still outperforms the two SOTA baselines, DoRA and PiSSA.

---

### Official Review · Reviewer_Qyxg · 2025-07-06

**Clarity:** 3
**Significance:** 3
**Originality:** 4
**Rating:** 4
**Confidence:** 1

**Summary:**

This paper proposes StelLA,  a geometry-aware extension of LoRA that uses decomposition **USV** instead of traditional **BA**. During the training process, StelLA constrains **U** and **V** to lie on the Stiefel manifold. A significant advantage of stelLA is flexibility.  It decouples geometric constraints from the choice of optimizer. The authors conduct experiments across multiple domains, demonstrating the superiority of  StelLA over vanilla LoRA and other variants.

**Questions:**

(1) Compared to traditional optimizers, StelLA introduces a significant amount of additional computation. Theoretically and practically, how much would the proposed method slow down the model training speed?

(2) By adding strict constraints during training, could the convergence speed become slow or even fail to converge? Convergence speed is already a pain point for LoRA, and modification in this direction will further reduce the convergence speed.

(3) Is there potential for algorithmic or hardware-level optimizations to speed up Stella?

(4) I expect to get some intuitive explanations: Why does applying SVD decomposition with orthogonal constraints during training improve model performance? Could this operation be merely playing a similar role to simple regularization?

(5) Why are the rank settings different in each scenario? The choices lack justification.

**Ethical Concerns:**

["NO or VERY MINOR ethics concerns only"]

**Final Justification:**

The main reason I raised the score is that the authors' response addresses my concerns. In the rebuttal, the authors elaborated on the training overhead and explained the value of maintaining matrix orthogonality.

The reason I assigned a low confidence score is that I am concerned about whether this line of work has strong practical value. In practice, imposing constraints on the training matrix often has a negative impact on convergence (although the authors claimed otherwise). In addition, the improvement in performance may not necessarily come from the decomposition of direction and magnitude as claimed by the authors.

**Limitations:**

The limitations section is missing.

**Paper Formatting Concerns:**

References should be in numerical form, such as [1].

**Quality:**

3

**Strengths And Weaknesses:**

Strengths

S1: It is an interesting and well-motivated idea to use a three-factor decomposition **USV** and constrains **U** and **V** to lie on the Stiefel manifold.

S2: The extensive experiments, covering multiple scenarios, demonstrate the effectiveness of the proposed StelLA.

S3: The paper is well-written, with a reasonable structure and a clear algorithm flow, making it easy for readers to understand the work presented.


Weaknesses

W1: The experiment only compares the number of trainable parameters, but lacks comparison of computational overhead or training time.

W2: Will there be a major drawback to computational complexity? Will the training cost be significantly increased due to the involvement of complex operations?

W3: References should be in numerical form according to NeurIPS 2025 Paper Formatting Instructions.

W4: There is no discussion of limitations.

W5: Why maintain the orthogonality of subspaces? This approach lacks motivation.

I have limited understanding of the manifold direction and find it difficult to determine whether the formulas in the article are accurate.

---

> ### Author Rebuttal · Authors · 2025-07-31
>
> Thanks for the detailed review and constructive comments.
> We have addressed all the comments and suggestions in the revised version of the manuscript.
> Below we clarify the questions and concerns in detail.
>
> > W1: Lack of comparison of computational overhead or training time.
>
> and
>
> > W2: Will there be a major drawback to computational complexity? Will the training cost be significantly increased due to the involvement of complex operations?
>
> and
>
> > Q1: Theoretically and practically, how much would the proposed method slow down the model training speed?
>
> and
>
> > Q3: Is there potential for algorithmic or hardware-level optimizations to speed up Stella?
>
> Thanks for the comments and questions regarding the computational overhead and training time, which is indeed an important aspect of our proposed method.
> We would like to point out that a detailed discussion of computational cost of Alg. 1 has already been provided in the appendix Sec. D.
> We will move this information from the appendix to the main text for better visibility.
>
> We summarize the key information in the appendix here:
> - The main computational overhead of Alg. 1 comes from the SVD operations in the polar retraction step, which can be reduced by using efficient CUDA-accelerated SVD implementation [e].
> - Training the LLaMA3-8B model with StelLA using Alg. 1 takes 6.5h for the commonsense benchmark, which is approximately 1.4x the time of LoRA (4.5h) on one H100 GPU. It can be further improved, e.g., by increasing parallelism with batch processing (details below), which reduces the training time from 6.5h to **5.2h**, about only 15% slower than vanilla LoRA.
>
> Detailed discussion:
>
> Theoretically, the computational complexity of the polar retraction step is $O(mn^2)$ for $m\times n$ shaped tall matrices ($m > n$) due to the SVD computation.
> But practically, there are efficient SVD algorithms for tall matrices that can reduce the computational cost significantly.
> Specifically, we used the `gesvda` solver [e] in StelLA, which is a CUDA-accelerated SVD implementation that can handle tall matrices efficiently.
> Below is a micro-benchmark of different SVD solvers for the matrix shapes used in StelLA for the commonsense benchmark on LLaMA3-8B.
> The results show that the `gesvda` solver is significantly faster than the default GPU solver in PyTorch.
>
> **Table: Runtime of different SVD solvers in PyTorch on H100 GPU**
>
> | Matrix Shape | Default GPU Solver (ms) | GESVDA Solver (ms) | Speedup of GESVDA |
> | ------------ | ----------------------- | ------------------ | ----------------- |
> | 4096x32      | 0.938                   | 0.641              | **1.46x**         |
> | 1024x32      | 0.795                   | 0.632              | **1.26x**         |
> | 14336x32     | 1.530                   | 0.654              | **2.34x**         |
>
>
> The speed could be further improved by using batch processing to increase parallelism.
> Specifically, we can stack all the low-rank matrices $U$ and $V$ with the same shape into a batch, and then perform the polar retraction step on the batch of matrices.
> This reduces the training time from 6.5h to **5.2h**, about 15% slower than vanilla LoRA.
> Below is a micro-benchmark of the SVD using batch processing on the matrix shapes used in StelLA for the commonsense benchmark on LLaMA3-8B (there are 192, 64, and 64 matrices with shapes 4096x32, 1024x32, and 14336x32, respectively).
> The results show that the batched SVD is significantly faster than the sequential one, and the polar retraction step is no longer the bottleneck of the training time.
>
> **Table: Runtime of SVD with/without batch processing**
>
> | Matrix Shape | Batch Size | GESVDA with Batch (ms) | GESVDA without Batch (ms) | Speedup of Batched Processing |
> | ------------ | ---------- | ---------------------- | ------------------------- | ----------------------------- |
> | 192x4096x32  | 192        | 5.80                   | 123.1                     | **22.22x**                    |
> | 64x1024x32   | 64         | 1.62                   | 40.4                      | **24.94x**                    |
> | 64x14336x32  | 64         | 2.89                   | 41.8                      | **14.46x**                    |
>
>
> > W3: References should be in numerical form according to NeurIPS 2025 Paper Formatting Instructions.
>
> Thanks for pointing this out; we have fixed the reference format in the revised manuscript.
>
> > W4: There is no discussion of limitations.
>
> We have discussed the limitations in appendix Sec. G, which we will move to the main text for better visibility.
>
> > W5: Why maintain the orthogonality of subspaces? This approach lacks motivation.
>
> and
>
> > Q4: I expect to get some intuitive explanations: Why does applying SVD decomposition with orthogonal constraints during training improve model performance? Could this operation be merely playing a similar role to simple regularization?
>
> The columns of the orthogonal matrix form an orthonormal basis for the subspace, which represents the "direction" of the subspace and is decoupled with the "magnitude" of the representation.
> Maintaining orthogonality is to ensure that the learning focuses on optimizing the "direction" of the subspace rather than their "magnitudes".
> Specifically in StelLA, $V$ represents the "direction" of the input subspace, and $U$ represents the "direction" of the output subspace, and $S$ represents the "magnitude" of the adapter.
>
> The separation of direction and magnitude is a useful regularization in many areas of machine learning, such as in weight normalization [a] (separates direction and magnitude of network weights), CosFace [b] (normalize both features and class representative vector to focus on learning directions), and Grassmann Class Representation [c] (represent class prototype by subspaces, which ignores the magnitude).
> Research in these areas shows that decoupling direction and magnitude can improve the performance of the model.
> Recently, DoRA [d] also shows that this separation also benefits low-rank adaptation.
> As such, we investigated whether explicitly optimizing the direction of subspaces during low-rank training can further improve LoRA's performance.
> Our empirical results confirm that preserving orthogonality throughout training can indeed lead to performance gains.
>
> As reviewer CC7z pointed out, maintaining U and V on the Stiefel manifold is also beneficial for certain downstream analyses, e.g. adversarial robustness.
>
> We will enhance the discussion of the motivation for maintaining orthogonality in the revised version of the manuscript.
>
>
> > Q2: By adding strict constraints during training, could the convergence speed become slow or even fail to converge? Convergence speed is already a pain point for LoRA, and modification in this direction will further reduce the convergence speed.
>
> We didn't observe any convergence issues in our experiments.
> We used the same number of training epochs for both StelLA and all baselines.
> At the end of training, StelLA achieves better performance than other methods, which indicates that the added constraints do not hinder convergence but instead contribute to more effective optimization.
>
> Generally speaking, literature on adding orthogonal constraints [c, f, g, h, i] in neural networks shows that the convergence speed is improved with the orthogonality constraints.
>
> We will add comparison of convergence plots to the revised manuscript.
>
> > Q5: Why are the rank settings different in each scenario? The choices lack justification.
>
> For each learning task, we carefully followed the same setting as in previous work for fair comparison.
> For example, for the commonsense reasoning benchmark, we followed the setting in DoRA (L201), and for the math and code generation benchmarks, we followed the setting in PiSSA (L224).
>
> > Difficult to determine whether the formulas in the article are accurate.
>
> Thanks for raising this concern.
> We provided an illustrative example in the appendix Sec. A, where a two-dimensional case is used to explain the concepts related to Stiefel manifold and geometric optimization.
> The equations are simplified in two-dimensional space, hoping to help sanity checking the workflow of Alg. 1.
>
> - [a] Weight normalization: A simple reparameterization to accelerate training of deep neural networks, NIPS 2016.
> - [b] Cosface: Large margin cosine loss for deep face recognition, CVPR 2018.
> - [c] Get the Best of Both Worlds: Improving Accuracy and Transferability by Grassmann Class Representation, ICCV 2023.
> - [d] DoRA: Weight-Decomposed Low-Rank Adaptation, ICML 2024.
> - [e] https://docs.pytorch.org/docs/stable/generated/torch.linalg.svd.html
> - [f] Unitary Evolution Recurrent Neural Networks, ICML 2016.
> - [g] All you need is beyond a good init: Exploring better solution for training extremely
>  deep convolutional neural networks with orthonormality and modulation. CVPR 2017
> - [h] Can we gain more from orthogonality regularizations in training deep networks? NeurIPS 2018.
> - [i] Orthogonal Convolutional Neural Networks, CVPR 2020.

---

> > ### Comment · Reviewer_Qyxg · 2025-08-05
> >
> > Thank you for your detailed response, which has resolved most of my confusion.
> >
> > I still have one suggestion.I look forward to more detailed explanations on the practicality of the proposed method in industrial scenarios. From the current version, it appears that the improvement in experimental results is not as significant, especially for the text-to-image generation task. However, the resources required for computation and the difficulty of implementation have significantly increased.

---

> > > ### Author Response · Authors · 2025-08-05
> > > **Practicality of StelLA**
> > >
> > > We appreciate the reviewer’s follow-up question and are pleased that our previous response has addressed most of the concerns. Below, we elaborate on why StelLA is practical in industrial scenarios, highlighting its ease of use and implementation, strong empirical performance, and efficiency—even though the underlying theory is sophisticated.
> > >
> > > * **Ease of use and implementation**. StelLA is designed with usability and modularity in mind. We integrated StelLA into Huggingface's `peft` [j] library (L155-156), which is the de facto standard codebase for the state-of-the-art parameter-efficient fine-tuning methods. This integration makes StelLA as easy to use as existing methods such as LoRA. To demonstrate this, below we sketch the PyTorch pseudo-code for training StelLA.
> > >
> > > ```python
> > > from transformers import AutoModelForCausalLM, TrainingArguments, ...
> > > from peft import StellaConfig, get_peft_model, ...
> > >
> > > # Load base model
> > > model = AutoModelForCausalLM.from_pretrained(base_model, ...)
> > >
> > > # StelLA config
> > > config = StellaConfig(r=stella_rank, alpha=stella_alpha, target_modules=...)
> > >
> > > # Equip the model with StelLA adapters
> > > model = get_peft_model(model, config)
> > >
> > > # Setup trainer
> > > trainer = StellaTrainer(
> > >     model=model,
> > >     args=TrainingArguments(learning_rate=0.0005, optim="adamw_torch", ...)
> > >     train_dataset=...)
> > >
> > > # Train
> > > trainer.train()
> > >
> > > # Save
> > > model.save_pretrained(output_dir)
> > > ```
> > >
> > > This code follows the standard boilerplate for training a peft model, with only one notable addition: we use the custom `StellaTrainer` instead of the default `transformers.Trainer`.
> > > This trainer implements Alg. 1 using hooks, resulting in modular and clean code, as illustrated below:
> > >
> > > ```python
> > > from transformers import Trainer
> > >
> > > # StellaTrainer is a subclass of Trainer
> > > class StellaTrainer(Trainer):
> > >     def create_optimizer_and_scheduler(self, num_training_steps: int):
> > >         super().create_optimizer_and_scheduler(num_training_steps)
> > >         # Register optimizer pre-step and post-step hooks
> > >         self.optimizer.register_step_pre_hook(self._optim_step_pre_hook)
> > >         self.optimizer.register_step_post_hook(self._optim_step_post_hook)
> > >
> > >     def _optim_step_pre_hook(self, optimizer, args, kwargs):
> > >         # Implement line 5 of Alg. 1.
> > >         # Optionally, group all U and V parameters according to their shapes for speeding up
> > >         # This function is automatically called before optimizer.step()
> > >         ...
> > >
> > >     def _optim_step_post_hook(self, optimizer, args, kwargs):
> > >         # Implement lines 7-8 of Alg. 1.
> > >         # Optionally, group all U and V parameters according to their shapes for speeding up
> > >         # This function is automatically called after optimizer.step()
> > >         ...
> > > ```
> > >
> > > This design demonstrates several key strengths:
> > >     (a). **Simplicity**: StelLA requires minimal code modifications to existing training pipelines.
> > >     (b). **Modularity**: Hooks are cleanly separated and compatible with any base Euclidean optimizer.
> > >     (c). **Extensibility**: Speedups (e.g., via batched SVDs) or variants of the algorithm can be implemented with minimal changes.
> > >
> > > * **Strong Empirical Performance**. Extensive experiments on diverse tasks and models have demonstrated StelLA's strong performance, robustness, and versatility compared to other LoRA variants. For the text-to-image generation task, StelLA ranks the top in 7 out of 10 metrics in the SD 1.5 setting and ranks the top in 5 out of 10 metrics in the SD 2.0 setting. These results clearly outperform strong baselines, including recent state-of-the-art methods such as DoRA and PiSSA.
> > > * **Efficiency**. StelLA can be efficiently implemented, as detailed in the rebuttal. There is still room for further improvement, such as integrating it into DeepSpeed [k]. In DeepSpeed, the optimizer flattens and concatenates all parameters in the model, which improves the memory access pattern. This technique can likely enhance the efficiency of StelLA as well.
> > >
> > > We hope this clarifies the practical advantages of StelLA, and we thank the reviewer once again for the thoughtful feedback.
> > >
> > >
> > > [j] https://github.com/huggingface/peft
> > >
> > > [k] https://github.com/deepspeedai/DeepSpeed

---

> > > ### Author Response · Authors · 2025-08-07
> > > **Follow-up**
> > >
> > > Dear reviewer Qyxg, thank you again for your time and feedback. We'd be happy to clarify any remaining concerns or questions you might have. Please feel free to let us know if there's anything we can expand on.

---

### Comment · Area_Chair_E2yJ · 2025-08-06
**Reminder: Participate in Author Discussions Before Submitting “Mandatory Acknowledgement”**

Dear Reviewers,

Thank you for your contributions during the reviewer–author discussion period. If you have not yet engaged with the authors, please do so at your earliest convenience.

As per NeurIPS 2025 policy, reviewers must participate in discussions with authors before submitting the “Mandatory Acknowledgement.” If you have already submitted the acknowledgement without engaging in discussion, we kindly request that you participate in the discussion now.

Your input at this stage is essential to ensuring a fair and thorough review process.
Thank you again for your dedication to the review process.

Best regards,
AC

---

### Decision · Program_Chairs · 2025-09-17

**Decision:**

Accept (spotlight)

**Comment:**

This paper introduces StelLA, a principled extension of LoRA that leverages a three-factor decomposition with orthonormal subspaces constrained on the Stiefel manifold. The work empirically demonstrates that these constraints ensures geometric consistency and stability of adaptation while remaining compatible with standard optimizers. Reviewers highlighted several strengths: (i) a clear and well-motivated novelty in constraining LoRA factors on the Stiefel manifold to address a limitation of prior works, (ii) a well-written, accessible presentation with sound algorithmic design, and (iii) experiments across diverse tasks (classification, language modeling, and text-to-image generation). Overall, the paper advances the field of parameter-efficient fine-tuning, and given its relevance to both research and practice, I recommend acceptance.